# Tumor diversity and the trade-off between universal cancer tasks

Jean Hausser [1], Pablo Szekely[1], Noam Bar[1], Anat Zimmer[1], Hila Sheftel[1], Carlos Caldas [2,3]* & Uri Alon [1]*

Recent advances have enabled powerful methods to sort tumors into prognosis and treatment groups. We are still missing, however, a general theoretical framework to understand the vast diversity of tumor gene expression and mutations. Here we present a framework based on multi-task evolution theory, using the fact that tumors need to perform multiple tasks that contribute to their fitness. We find that trade-offs between tasks constrain tumor gene-expression to a continuum bounded by a polyhedron whose vertices are gene-expression profiles, each specializing in one task. We find five universal cancer tasks across tissue-types: cell-division, biomass and energy, lipogenesis, immune-interaction and invasion and tissue-remodeling. Tumors that specialize in a task are sensitive to drugs that interfere with this task. Driver, but not passenger, mutations tune gene-expression towards specialization in specific tasks. This approach can integrate additional types of molecular data into a framework of tumor diversity grounded in evolutionary theory.

[1] Department of Molecular Cell Biology, Weizmann Institute of Science, 76100 Rehovot, Israel. [2] Department of Oncology and Cancer Research UK Cambridge Institute, University of Cambridge, Cambridge CB2 0RE, UK. [3] Breast Cancer Programme, Cancer Research UK Cambridge Cancer Centre, Cambridge CB2 0RE, UK. *email: carlos.caldas@cruk.cam.ac.uk; uri.alon@weizmann.ac.il

Tumors show diversity in genetic alterations, gene expression and drug sensitivities, presenting a major medical challenge. Tumors continually evolve[1–4] with driver mutations (single nucleotide variants (SNVs), copy number alterations (CNAs), and translocations) conferring a selective advantage. Yet tumors differ in which driver mutations they carry[5,6]. Understanding tumor diversity, comprehending the function of driver mutations in different contexts, and understanding why drugs affect some tumors and not others, are all pressing fundamental questions[7].

The growth in molecular data on tumors has driven the development of powerful algorithms to sort tumors into classes and clusters[8–10]. These algorithms attempt to maximally separate tumors and cluster them according to molecular criteria. Cluster membership can be correlated with drug response and prognostics to help guide and design treatment for individual patients based on their molecular profile[6,11–14].

While the ability to sort tumors is powerful and useful, there remains an open question of understanding, from a theoretical basis, why tumors vary in the way that they do. To address this, we apply a theory of multitask evolution to the case of cancer. We reasoned that multitask evolution may apply to cancer because cancer is a case of intense evolution inside the body that can play out over years, with generation times of cells that can be on the order of days[15–18]. Furthermore, cancer cell growth and survival is conditioned on fulfilling multiple tasks, including growth, stress resistance, interaction with the immune system and so forth[19]. Each task requires a different profile of gene expression—ribosomes for growth and stress proteins for survival. Presumably no tumor can be optimal at all tasks at once, because cells can only make a limited amount of protein per unit biomass, and proteins for different functions can interfere with each other. Thus, cell communities that optimally manage the trade-off relevant for their particular niche in the body will outgrow and out-survive cells that are suboptimal.

Trade-offs are often found when resources (nutrients, time, and space) are limited and have been well-studied in evolutionary ecology[20]. A well-known example of trade-offs is found in bacteria: cells that grow faster are more sensitive to stress and antibiotics[21]. There is selection on cells that grow in a challenging environment to express survival genes, which comes at the expense of growth genes[22,23]. A similar trade-off is found in cancer cells[24]. For example, cancer cells exposed to hypoxia can survive by invading the tissue surrounding the tumor[25,26] which can come at the cost of a reduced proliferative activity[27,28].

But growth and survival are only two of the possible tasks that affect tumor cell fitness. How can we detect and understand trade-offs between three and more tasks? How can we identify the tasks without assuming what the tasks are a priori? Going beyond a trade-off between two tasks requires special approaches that can detect the impact of multiple simultaneous tasks.

To address the question of trade-offs in tumors between multiple tasks, we apply multitask evolutionary theory, known as Pareto task inference (ParTI), to cancer. We use ParTI to (i) identify trade-offs between five universal tasks shared across cancer types, (ii) show that tumors that specialize in a task are differentially sensitive to drugs that disrupt that task, and (iii) demonstrate that each driver mutation moves gene expression towards specialization in specific tasks. This suggests a picture of tumor diversity based on multitask evolution. Thus, our goal is not to separate tumors, as is already done well by existing algorithms, but to add to our understanding of which evolutionary trade-offs lead to the observed variation between tumors, and to explain drug sensitivity and driver mutations in terms of the concept of specialist/generalist tumors at different tasks.

## Results

**Tumor gene expression profiles fall on polyhedra.** The starting point for ParTI is that a tumor needs to perform multiple tasks in order to thrive[19], but that these tasks are not known a priori. ParTI further assumes that no tumor can be optimal at all tasks at once, leading to a fundamental trade-off. Due to the intense competition and high turnover of cells in a tumor, natural selection is expected to be strong[15–18]. Cells with suboptimal gene expression will lose the competition to cells with gene expression that is more optimal given the trade-offs. It thus makes sense to apply Pareto optimality theory to the tumor situation. The theory predicts that such trade-offs lead to a characteristic pattern: gene expression, averaged over all of the cell types in the tumor, is arranged within a polyhedron in gene expression space, a geometric structure with flat sides and sharp vertices[23]. For example, a polyhedron with two vertices is a line (Fig. 1a). A polyhedron with three vertices is a triangle (Fig. 1b). Four vertices describe a tetrahedron (Fig. 1c). The vertices of the polyhedron, called archetypes, are profiles optimal for one of the tasks (Fig. 1a). Specialist tumors at a task lie close to a vertex, and generalists lie in the middle of the polyhedron (Fig. 1b). The fundamental reason for the polyhedron is that it is the shape that encloses all points closest to the archetypes. Tumors outside the polyhedron are suboptimal, and will not be selected.

Thus, finding polyhedral structure in data allows one to infer the number and nature of the tasks. Such polyhedral structures, tasks and trade-offs were found in several contexts including bacterial and eukaryotic cell gene expression, animal morphology (Fig. 1a–c)[23,29–32] and in a preliminary analysis of breast cancer[33] and Wilms' tumours[34].

To test whether human tumor transcriptomes fall on low-dimensional polyhedra, we analyzed the transcriptomes of primary tumor samples from TCGA[35] and Metabric[5,36] (normal samples were removed). We used the ParTI software package[33] which fits lines, triangles, tetrahedra and so on to data, finds the best fit polyhedron. The statistical significance of fitting a polyhedron to the data is assessed by the $t$-ratio test[23,33] that compares how well the data fills the polyhedron compared with the randomized datasets (inset of Fig. 1d). ParTI thus infers the number of archetypes and their position in gene expression space. We tested the 15 cancer types that have at least 250 primary tumor samples. We find that polyhedra with 3–5 archetypes describe gene expression of six cancer types, including breast, colon, thyroid, bladder, low-grade glioma, and liver (Fig. 1d, Supplementary Fig. 1A, False discovery rates (FDR) < 10%, $p <$ 0.001 to $p = 0.009$ at $t$-ratio test), with two more showing borderline significance (lung, $p = 0.01$ and head and neck, $p = 0.02$, $t$-ratio test).

The seven other cancer types (including kidney renal clear cell carcinoma and ovarian cancer) appear as clouds in gene expression space without detectable vertices; possible reasons include having primarily one task and hence no strong trade-offs, having too many tasks and thus too many vertices to resolve, or data heterogeneity not currently understood.

We find that the archetypes (vertices) of the polyhedra for different cancer types are similar to each other in terms of gene expression (Supplementary Fig. 1I–J). We therefore hypothesized that tumors from different cancer types face similar trade-offs. To test this, we pooled the 3180 primary tumors from the six cancer types, after correcting gene expression profiles for tissue identity (normalizing each sample by the mean expression profile of its cancer type, see "Methods" section). We find that transcriptomes of the tumors vary in a continuum inside a polyhedron bounded by five archetypes ($p = 0.002$ at $t$-ratio test, Fig. 2a). Tumors from different cancer types are spread widely within the polyhedron (Fig. 2a, Supplementary Fig. 2A, B), and are found close to 3–5 of

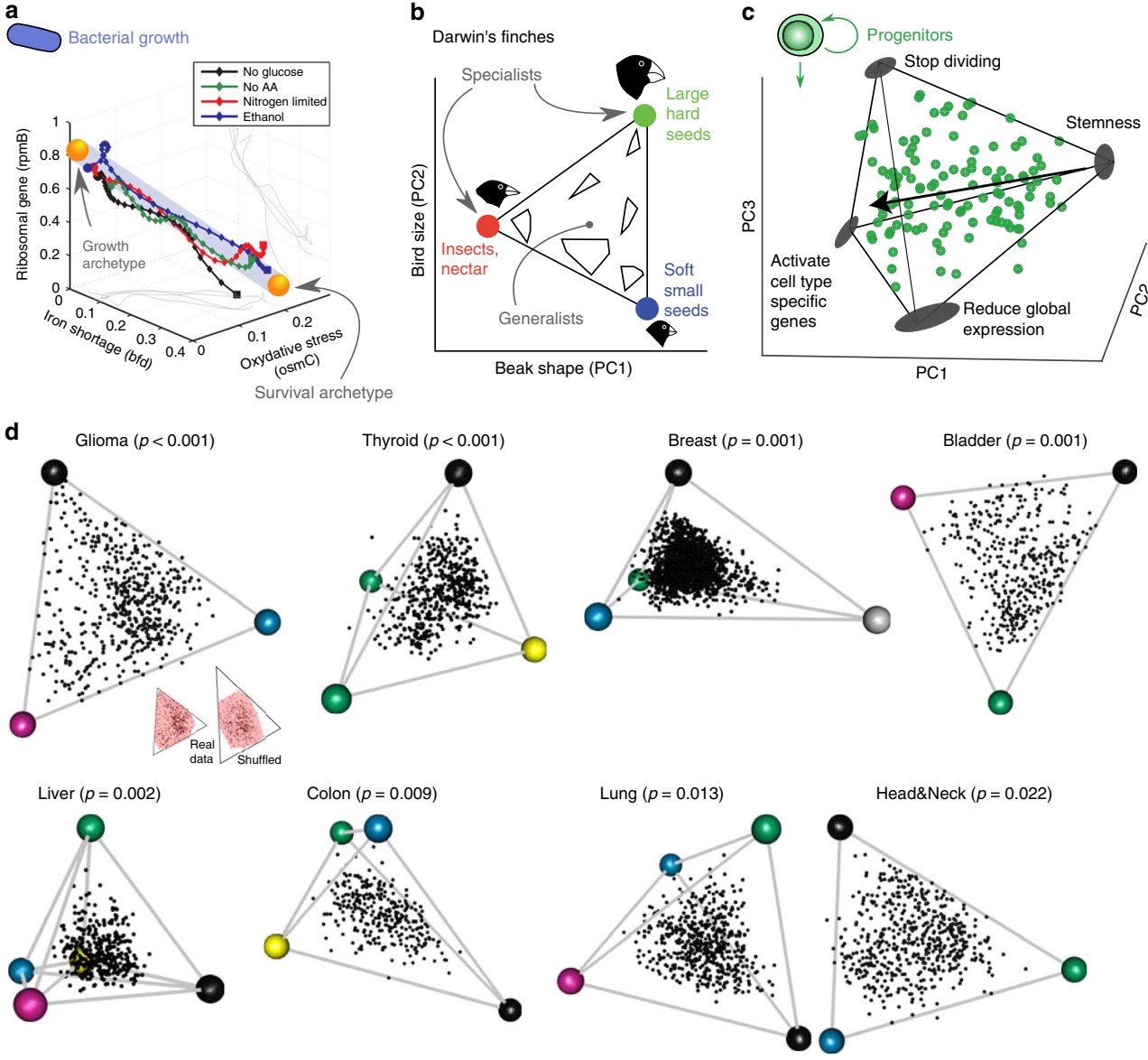

**Fig. 1** Trade-offs between tasks leads to phenotypes arranged in a polyhedron, whose vertices are archetypes that specialize in one task, in situations ranging from bacterial gene expression to animal morphology to cancer. **a** 90% of the variation in *E. coli* gene expression falls on a line due to a trade-off between tasks of growth and survival. Axes are percent of total promoter activity. **b** Morphology of Darwin's ground-finch species falls on a triangle. Specialists in different diets are found near the three archetypes, and generalist are near the center of the triangle. A and B adapted with permission from AAAS from Shoval et al.[23]. **c** Single-cell gene expression of mouse intestinal progenitor cells fall on a tetrahedron, shown in principal components (PC) space. The four archetypal gene expression profiles correspond to fundamental progenitor cell tasks. Adapted from Korem et al.[29]. **d** Tumor gene expression profiles of eight cancer types fall on polyhedra. Individual tumors (dots) plotted in the space spanned by the first three gene expression PCs (TCGA, breast cancer from Metabric). Archetype (colored dots) number and position were inferred using ParTI. Inset: shuffled data has a convex hull (CH, pink) that fills less of the minimal enclosing (ME) triangle than the real data. The ratio of the CH area (or volume) and ME triangle area (or tetrahedron volume) was used to compute statistical significance.

the archetypes depending on the cancer type (Table 1, Supplementary Fig. 2A-D). Tumors are not found in the immediate vicinity of archetypes for statistical reasons (Supplementary Note 1, Supplementary Fig. 2E).

To infer the tasks performed by these five universal archetypes, we analyzed which pathways and functional gene groups are expressed highest in the tumors closest to a given archetype, using MSigDB[37] (Supplementary Data 1, FDR < 10% at Mann–Whitney test, using leave-one-out controls). We also determined which clinical properties are frequent among the tumors closest to a given archetype compared with other tumors in the dataset (Supplementary Data 2–3, FDR < 10% at Mann–Whitney test).

We find clear tasks for each of the five archetypes (Table 2, Supplementary Fig. 2F). The five tasks are: cell division, biomass and energy production, lipogenesis, immune interaction, and invasion and tissue remodeling. The tasks match the hallmarks of cancer defined by Hanahan and Weinberg[19] (Fig. 2b). A given hallmark can contribute to one or more tasks.

Tumors from patients with higher number of invaded lymphnodes are found near the invasion and tissue remodeling archetype ($p < 10^{-3}$, Mann–Whitney test, Supplementary Data 3). Tumors with highest histological grade (which corresponds to poor tissue differentiation) are found near the immune interaction archetype ($p = 10^{-6}$, hypergeometric test, Supplementary

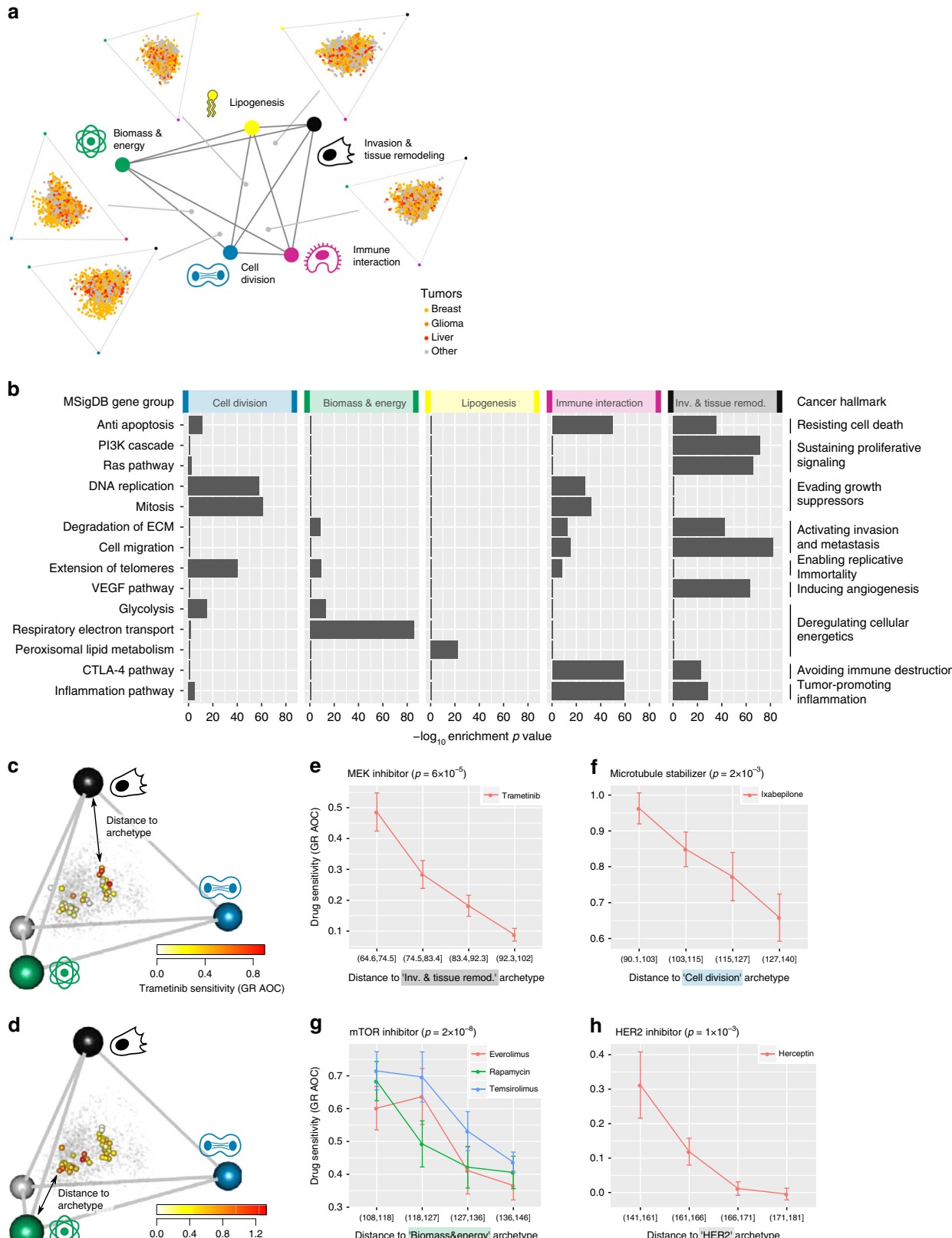

Data 2). Tumors at a relatively early stage (Stage II) are found near the cell division, biomass and energy and lipogenesis archetypes ($p < 0.003$, hypergeometric test) whereas late-stage tumors (Stage III) are found near the immune interaction and invasion and tissue remodeling archetypes ($p < 10^{-5}$, hypergeometric test, Supplementary Data 2).

The tasks for each tissue type are indicated in Table 1, and color coded on the archetypes of Fig. 1d. Each tissue type seems to show a trade-off between 3–5 universal tasks (Table 1, Supplementary Fig. 2D). Breast cancer shows three universal archetypes (division, invasion, biomass and energy) and a fourth one enriched with HER2-positive tumors ($p < 10^{-8}$,

**Fig. 2** Tumors face trade-offs between five universal cancer tasks. **a** Gene expression profiles of tumors from six cancer types fall in a continuum bounded by five archetypes, with indicated tasks. Tumors are shown projected on polyhedron faces (light orange—breast, dark orange—glioma, red—liver, gray—other tissue). Archetype number and position (colored dots) were inferred using ParTI. **b** The five cancer tasks represent hallmarks of cancer[19]. For each cancer hallmark, one to three representative MSigDB gene groups were chosen. *P*-value are for enrichment in the 5% of the tumors closest to each cancer archetype (Mann–Whitney test, $n = 159$ tumors). For visualization, gene groups that are not significant at FDR < 0.1 or are not maximally enriched at a given archetype are set to 1. **c–d** Breast cancer (BC) cell-line gene expression[14], projected on the BC tumor tetrahedron. Color: sensitivity of cell lines to indicated drug. Grey dots: breast tumors. **e–h** Sensitivity to indicated drug, defined as area under curve (AOC) of growth rate (GR) dose response, as function of Euclidean distance of cell-line gene expression to indicated archetype[14,38]. Error bars represent SE. *P*-values from the Mann–Whitney test ($n = 14$ cell lines).

**Table 1 Universal cancer tasks found in each cancer type.**

|  | Glioma | Thyroid | Breast | Bladder | Liver | Colon |
|---|---|---|---|---|---|---|
| cell division | X |  | X |  | X | X |
| biomass & energy |  | X | X | X | X | X |
| lipogenesis |  | X |  |  | X | X |
| immune interaction | X |  |  | X | X |  |
| inv. & tissue remod. | X | X | X | X | X | X |

X indicates the significant overlap between cancer archetype and universal archetype in the set of genes whose expression is enriched in tumors closest to archetype (see "Methods" section)

hypergeometric test, Supplementary Data 2), which seems to be tissue specific (Supplementary Fig. 2D).

**Polyhedra do not result from averaging different cell types**. In interpreting the task of the archetypes, one concern is that we use data averaged over all cells in the tumor. The different archetypes could represent pure cell types (immune cells, stromal cells, malignant cells,…) rather than tasks. Such a situation would also result in polyhedra: if tumors are weighted average of cell types, they fall on a polyhedron with pure cell-type profiles at the vertices.

However, the data is inconsistent with the hypothesis that archetypes represent individual cell types. If archetypes represented pure cell types, tumors should fall on polyhedra in *linear* gene expression space. We find no significant polyhedra in linear gene expression space, only in log gene expression space (Supplementary Fig. 1A, B). Furthermore, if archetypes represented pure cell types, tumor purity should be lowest close to all the archetypes that represent non-cancer cell types and highest close to the one archetype that represents cancer cells. We find that purity is significantly elevated at multiple archetypes in glioma ($p < 10^{-10}$, Mann–Whitney test, Supplementary Fig. 1C), thyroid cancer ($p < 10^{-10}$), hepatocarcinoma ($p < 10^{-8}$), and colon cancer ($p < 10^{-8}$) (Supplementary Data 3). These observations suggest that archetypes do not represent pure cell types.

The identified archetypes are robust to variations in tumor purity and clonal heterogeneity (Supplementary Fig. 1D–H, Supplementary Note 2).

**The sensitivity of tumors to drugs depends on tasks**. To further test the hypothesis that tumors face trade-offs between conflicting tasks, we sought a way to test whether tumors near an archetype depend on its task more than other tumors. For this purpose, we employ the drug sensitivity of different tumors, reasoning that tumors near an archetype will be most sensitive to drugs which specifically disrupt that task.

We used data from Heiser et al. who assessed the sensitivity of 49 human breast cancer cell lines to a panel of 77 drugs[14]. This dataset includes growth rate, overcoming a limitation of larger

datasets in which apparent drug sensitivities cannot be corrected for growth rate effects[38]. We determine the position of these cell lines relative to the breast cancer archetypes by projecting the transcriptome of the cell lines onto the gene expression space defined by breast tumors (Fig. 2c, d, see "Methods" section).

We find that cell lines closest to the invasion and tissue remodeling archetype are sensitive to trametinib, an inhibitor of the Ras pathway which is upregulated in tumors close to the invasion and tissue remodeling archetype (Fig. 2e, Table 2). Similarly, cell lines closest to the cell division archetype are sensitive to ixabepilone which stabilizes microtubules, and thus targets mitosis (Fig. 2f). Cell lines near the biomass and energy archetype are most sensitive to drugs which inhibit mTOR (Fig. 2g), a controller of cell growth[39]. Finally, cell lines close to the breast cancer-specific HER2 archetype (tumors that overexpresses the erbB-2 receptor) are sensitive to herceptin, an erbB-2 inhibitor (Fig. 2h). This differential sensitivity to drugs supports the hypothesis that tumors close to archetypes are task specialists.

**Driver mutations specialize tumors in specific tasks**. We next asked how genetic alterations in tumors fit into the trade-off picture. We computed the mean effect of each genetic alteration, a vector that describes how this alteration shifts gene expression (the difference in gene expression between tumors with and without the alteration, schematically shown in Fig. 3a). We compared this effect vector with the polyhedron for each cancer type. There are two possible situations: the effect vector can align with the polyhedron, or instead can point away from the polyhedron. To visualize this, if the front were a triangle, the effect vector could lie on the same plane as the triangle (have a small angle with the plane), or could point away from the plane (have a large angle) (Fig. 3a). Importantly, shuffled controls, in which the mutation data is shuffled between cancers, typically have effect vectors that point away, (angle = 60–80°) because the polyhedron explains only a fraction (20–40%) of the variation in the data.

Strikingly, for five cancer types, the effect vectors of driver single nucleotide variants (SNVs) align with the polyhedron much more closely than expected from shuffled data: glioma ($p = 10^{-4}$, shuffling test, Fig. 3b), thyroid cancer ($p = 10^{-3}$), breast cancer ($p < 10^{-8}$, Fig. 3c), bladder cancer ($p = 0.02$), and colon cancer ($p = 5 \times 10^{-3}$, Supplementary Fig. 3A). Drivers are also much more aligned than non-driver cancer genes and passenger SNVs collected from[6,36,40–42]. The latter are as aligned as shuffled data (Supplementary Fig. 3A).

We further find that driver SNVs move gene expression towards specific archetypes. For example, *IDH1*, a strong driver in glioma, shifts gene expression towards the cell division archetype (Fig. 3d). In breast cancer, the common *TP53* mutation is the most aligned with the front. It points directly towards one archetype, cell division (angle to archetype = 18°, mutation enriched 2.6-fold in the 5% of tumor closest to archetype, $p < 10^{-16}$, hypergeometric test, Fig. 3e). Mutations in *TP53* in breast cancer and *IDH1* in glioma thus coordinate gene expression towards specializing in the cell-division task. Another breast

**Table 2 Each cancer task is characterized by specific gene expression programs and clinical properties.**

| | Task | Highly expressed genes | Clinical properties | Drug sensitivities |
|---|---|---|---|---|
| | Cell division | cell cycle, M phase, DNA replication, telomere extension | Stage II, long recurrence free survival, high #CNAs, triple negative breast cancer of high cellularity | ixabepilone *stabilizes microtubules* |
| | Biomass & energy | ribosome (mitochondrial and cytosolic), respiration, glycolysis, proteasome | Stage T2 | rapamycin, temsirolimus, everolimus *mTOR inhibitors* |
| | Lipogenesis | de novo lipid synthesis, glycosylphosphatidylinositol biosynthesis, peroxisome lipid metabolism | Stage II, low grade (well differentiated tissue), high % tumor nuclei, older patients, ER+/PR+ breast cancer | not tested *archetype not found in breast cancer* |
| | Immune interaction | Allograft rejection, cytotoxicity, inflammation, IFNγ signaling, PD-1 and CTLA-4 signaling | Stage III, high #SNVs, high grade (poor differentiation) | not tested *immune checkpoint therapy not tested* |
| | Invasion & tissue remodeling | ECM remodeling, cell migration, angiogenesis, signaling (Hh, IGF, FGF, Wnt, TGFβ, Ras), lectin pathway, neurogenesis | Stage IIIC, lymphatic invasion, high % stroma | trametinib *MEK1/2 inhibitor* |

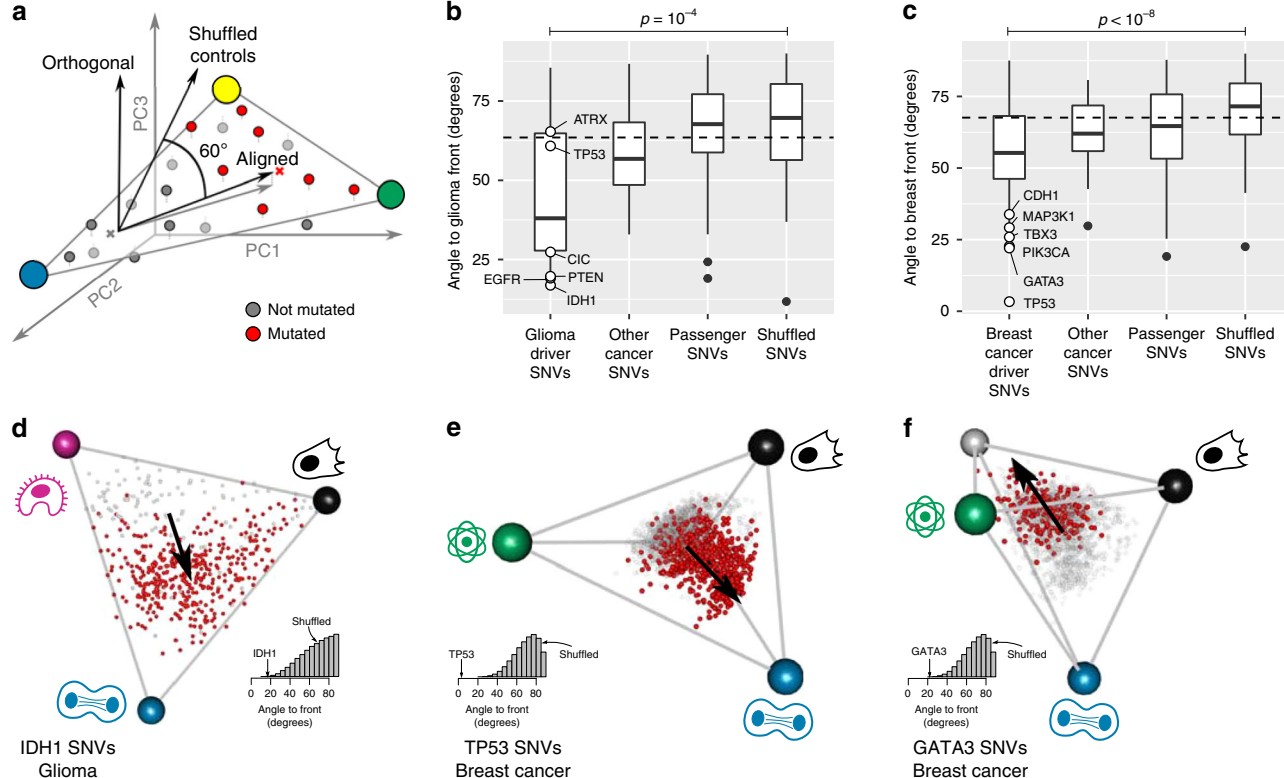

**Fig. 3** Driver mutations push tumor expression towards specialization in specific tasks. **a** The effect of a mutation on gene expression is defined as the vector (grey arrow) connecting the centroids of tumors without (gray) and with (red) the mutation (schematic). Alignment of mutation effect vector to the front is defined by the angle of the effect vector to the subspace defined by the front. Shuffled controls show a 62–75° angle with the front depending on cancer type. **b**–**c** In glioma (**b**) and breast cancer (**c**), driver mutations are better aligned to the polyhedron formed by the archetypes than passenger mutations and random mutations. White dots: six most frequent driver SNVs in that cancer type. Center line, median; box limits, upper and lower quartiles; whiskers, 1.5× interquartile range; black dots, outliers. *P*-values from the Mann–Whitney test ($n = 18$ driver mutations for glioma, $n = 53$ for breast). **d** In low-grade glioma, *IDH1* SNVs point approximately towards the cell division archetype. **e** In breast cancer, *TP53* SNVs point towards the cell division archetype. **f** In breast cancer, *GATA3* SNVs point towards the face defined by the lipogenesis, invasion and tissue remodeling and HER2 archetype.

cancer driver, *GATA3*, shifts gene expression towards the face defined by the lipogenic, *HER2* and invasion and tissue remodeling tasks and away from the cell division archetype ($p = 0.003$, hypergeometric test, Fig. 3f). The same conclusion is

found for all 21 drivers with significant alignment to the polyhedra (Supplementary Data 4 lists the driver SNVs and the archetypes they point to, FDR < 10% at mutation shuffling test). Thus, aligned driver mutations can be interpreted as knobs that

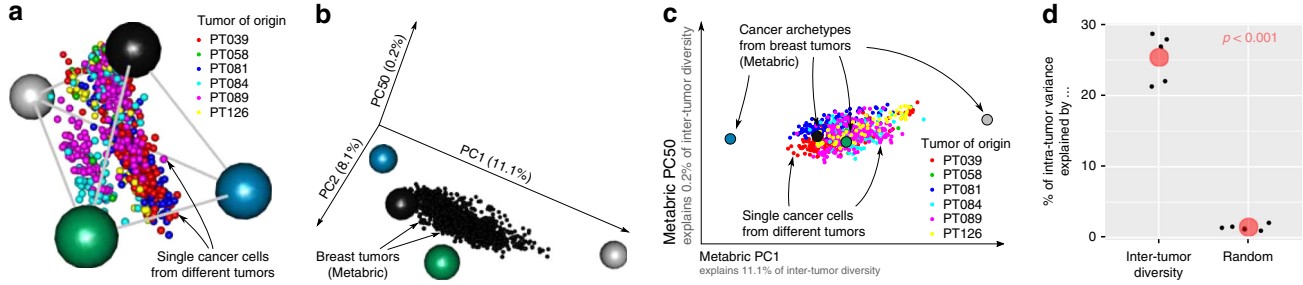

**Fig. 4** Cancer tasks partly explain intratumor heterogeneity. **a** Projecting single-cancer cells from different breast tumors onto the tetrahedron defined by metabric breast tumors shows that single-cancer cells from the same tumor can be generalists and specialists at different tasks. **b** Intertumor diversity varies mainly along the first two principal components (PC) of the metabric tumors: the first two PC explain 11.1 and 8.1% of the variance in gene expression between tumors whereas PC50 explains only 0.2% tumors thus fall on a plane within the volume defined by PC1, PC2, and PC50. **c** Projecting single cancer cells from different breast tumors onto PCs from metabric tumors shows that intratumor heterogeneity is aligned with intertumor diversity. **d** Intertumor diversity explains a significant fraction of intratumor variance. PCs computed on metabric tumors explain 25.4% of the variance explained by PCs computed on single cells. PCs computed on a shuffled version of the metabric data explain only 1.5% of the variance explained by single-cell PCs. Black dots represent different tumors. *P*-value from the T test. Data: gene expression in Metabric tumors from Curtis et al.[5], single-cell gene expression in six breast tumors from Karaayva et al.[47].

tune gene expression towards some tasks and away from others. Although the tasks are universal, the drivers that shift gene expression toward each task are often tissue specific.

We also analyzed copy number alterations (CNAs). CNAs show the same features found for SNVs above: driver CNAs have effect vectors that are aligned with the polyhedron (Supplementary Fig. 3B, C, Supplementary Note 3) and push gene expression to specialize in specific tasks (Supplementary Fig. 3D). For example, *PTEN* deletion in lower grade glioma points to the immune archetype; *MYC* amplification in breast cancer points to the cell division archetype (inferred tasks for 229 CNAs are listed in Supplementary Data 5, FDR < 10%).

**Single-cancer cells can be task specialists or generalists**. So far we considered intertumor diversity in gene expression. Our results bear the question of how intertumor diversity is supported by intratumor heterogeneity, the variation between single-cancer cells inside a tumor[43,44]. One possibility is that single cells specialize in different cancer tasks, so that the tumor composition in single-cell specialists sets the position of tumors relative to archetypes. This possibility is inconsistent with the data: if tumors are made of specialist cells, specialist tumors should be most homogeneous in specialist cells, whereas generalist tumors should be heterogeneous. But we find that specialist breast tumors can be both homogeneous and heterogeneous (Supplementary Fig. 4A), and that tumors that specialize in cell division are more heterogeneous than generalist tumors ($p = 1.6 \times 10^{-5}$, Mann–Whitney test, Supplementary Fig. 4A). Projecting single-cancer cells from different breast tumors on the space defined by breast cancer tasks confirms that single cells from individual tumors can be specialist or generalists at the different tasks (Fig. 4a).

In connecting intratumor heterogeneity with intertumor diversity, one prediction based on concepts from ecology[45,46] is called "evolution along lines of least genetic resistance"—the finding that the main axes of variations between individuals in a given species is aligned with variation between species in the same taxon. In the case of cancer, single cells and tumors may play the role of individuals and species so that variation between single cells in a tumor will align with variation between different tumors because of the shared tasks[23]. Single-cell transcriptomics data from breast tumors[47] verify this prediction: intertumor diversity explains 25.4% of intratumor heterogeneity in gene expression (Fig. 4b–d, Supplementary Fig. 4B, C). This alignment between intertumor diversity and intratumor heterogeneity is not expected

by chance because the space of gene expression has thousands of dimensions along which tumors and cells can vary. Accordingly, random directions in gene expression only explains 1.5% of intratumor variation ($p < 0.001$ at Student's *t* test, Fig. 4d, Supplementary Fig. 4B, C). The remaining 74.6% could be explained by factors such as stochasticity in single-cell gene expression, measurement error and neutral variation.

## Discussion

The goal of this study is to provide a framework to understand the diversity of tumors in terms of multitask evolution. We used ParTI to discover the number of tasks and their biological nature. The present work suggests trade-offs between five tasks. While the proliferation versus survival trade-off was already documented in cancer, our results suggest that there are other trade-offs that could be of comparable importance in shaping tumor gene expression. We suggest trade-offs between organizing metabolism to grow on glucose versus using lipids for growth, and between rapid cell division, immune evasion, and tissue remodeling. Testing these trade-offs could be the object of follow-up experiments aimed at measuring performance at the five tasks by quantifying DNA replication rate, lipid metabolism, protein synthesis rate, immuno-resistance, and invasion rate in different tumors. If trade-offs are at play one should find negative correlation between the different measures of performance.

This framework predicts that tumors that specialize in a task should be more sensitive to drugs that impair that task. We find evidence for this prediction (Fig. 2c–h). Future studies on specific tumors with more samples and accuracy can probably uncover additional archetypes. These can offer hypotheses for which drugs to use on which tumor, and which combinations might work for generalists that trade-off between multiple tasks.

Furthermore, we find that most driver mutations tune gene expression not in arbitrary directions in gene expression space, but instead towards specialization in specific tasks. Non-driver mutations have more arbitrary effects on gene expression, and do not point towards specific archetypes. This could provide an approach to detect driver mutations and differentiate them from non-driver mutations.

Note that our goal is not to discover new cancer subtypes: existing approaches to sort tumors are better for this purpose[8–10]. Such sorting approaches are typically not based on a theoretical evolutionary basis, but instead aim to maximize the difference between tumors according to molecular features. The point of the

present study is not to offer an improved way to sort tumors, but to provide a unifying explanatory framework to rationalize why tumors vary in the way that they do based on multitask evolution. From this point of view, it is gratifying that most of the tasks we find are well-known, and the sensitivity to drugs is easily understandable in terms of known mechanisms.

The five tasks match the ten hallmarks described by Hanahan and Weinberg[19]. Each task combines several hallmarks. For example, gene groups correspondng to the hallmarks of evasion of growth suppressors and enabling replicative immortality both support the cell division archetype. The archetype of immune interaction combines gene groups for the hallmarks of resisting cell death, avoiding immune destruction, and tumor-promoting inflammation (Fig. 2b).

One question raised by our findings is why tumors vary in their degree of specialization in the different tasks. Some of this variation is associated to tumor stage: early-stage tumors are found closer to the cell division, biomass and energy and lipogenesis archetypes while late tumors are more often found close to the immune interaction and invasion and tissue remodeling archetypes. This suggests that variations in the degree of specialization of tumors in different archetypes could stem from changes in the trade-offs during tumor progression, with proliferation being more important during early stages and survival during later stages. Other variables such as patient age, driving genetic alterations, and type of alteration (CNAs vs single nucleotide variations) are also associated with different archetypes (Table 2). Interestingly, different combinations of trade-offs are identified in different cancer types (Table 1). This suggests that factors such as the cell of origin or differences in the microenvironment of different host tissues could explain why different tumors specialize in different tasks. Untangling the causalities that underlie these variations could be the aim of future research.

There could be alternative interpretations to the observed associations between cancer tasks and driver mutations. For example, different cell types of origin may have different propensity to specialize in different tasks and acquire different mutations. At the moment, this specific explanation has limited support, at least in breast cancer: most breast cancers are thought to originate from luminal progenitors, so that most intertumor diversity cannot be explained by the cell of origin, except in rare tumors (claudin low). Nevertheless, clarifying the origin of the observed association between driver mutations and cancer tasks could be the object of future research, perhaps by comparing gene expression in genetically engineered mouse models with different driver mutations.

Given the genetic diversity of cancer, it is interesting to ask how multitask evolution handles the possibility that multiple gene expression profiles may maximize performance at a task. If different gene expression profiles optimize performance at the same task, there will be multiple archetypes for this task, each with its own gene expression profile. We do not observe this: the archetypes we find clearly differ in their tasks. However, comparing genes for a given task in different tumor types reveals that tasks have a tissue-specific flavor (Supplementary Data 6).

In summary, we suggest a framework for understanding tumor variation based on evolution under trade-offs between tasks. Tumor gene expression lies in a continuum in a polyhedron whose five vertices are archetypal expression programs for five tasks that recur in different cancer types. Tumors can be specialists at a task or generalists: specialists have gene expression close to a vertex and generalists lie in the middle of the polyhedron. We find support for the hypothesis that specialists in a task are more sensitive to drugs that disrupt that task. Driver mutations are often like knobs that tune gene expression towards specialization in specific tasks. This framework, if validated by

further research, offers a way to understand tumor variation in terms of task specialization.

## Methods

**TCGA data**. Gene expression data: we downloaded the RSEM-normalized HiSeqV2 gene expression data (log2 RPKMs) from the TCGA data portal. The TCGA data portal is now retired but the data can be retrieved from the Genomics Data Commons portal of the National Cancer Institute [https://gdc.cancer.gov/]. The starting point of our analyses was the 'genomicMatrix' files which contain expression levels for 20,530 human genes in 15 cancer types. We considered all cancer types with at least 250 primary tumor samples: bladder (TCGA disease code: BLCA)[48]; breast (TCGA disease code: BRCA)[49]; cervical (TCGA disease code: CESC)[50], colon (TCGA disease code: COAD)[51]; head and neck (TCGA disease code: HNSC)[52]; kidney (TCGA disease code: KIRC)[53]; lower grade glioma (TCGA disease code: LGG)[54]; liver (TCGA disease code: LIHC)[55]; lung adenocarcinoma (TCGA disease code: LUAD)[56]; lung squamous cell carcinoma (TCGA disease code: LUSC)[57]; ovarian cancer (TCGA disease code: OV)[58]; prostate (TCGA disease code: PRAD)[59]; stomach (TCGA disease code: STAD)[60]; thyroid (TCGA disease code: THCA)[61]; uterus (TCGA disease code: UCEC)[62].

SNVs and CNAs: from the TCGA data portal, we downloaded SNV calls, and CNA calls (gistic2 thresholded), as reported in 'genomicMatrix' files. For SNVs, we focused on genes mutated in at least 1% of the samples, to reduce the computational time and the memory requirements of our analyses. We also analyzed the CNAs which are prevalent. Since unlike mutations which can be present or absent, CNAs have 5 different values, we used an entropy measure, and analyzed the 1% of genes with highest entropy in their CNAs. For each gene, we determine the fraction of samples with (1) strong deletions $f_1$, (2) weak deletion $f_2$, (3) no detectable CNA $f_3$, (4) weak amplifications $f_4$, and (5) strong amplification $f_5$. We then computed the entropy of the corresponding distribution, $-\Sigma_i f_i \log f_i$. A gene whose copy number is never altered has entropy 0. Genes with highest entropy show the most frequent CNAs.

Clinical data: From the TCGA data portal, we also downloaded the clinical data reported in files named 'clinical_data'. Discrete and continuous clinical features require different statistical treatment in order to determine which clinical features are overrepresented among tumors close to individual archetypes. We thus separated clinical features into discrete and continuous features by manual examination.

**Metabric data**. For gene expression and CNAs, we downloaded the normalized microarray gene expression data (data_expression.txt), CNAs data (data_CNA.txt), and clinical data (data_clinical.txt) for the 1970 tumors of the Metabric cohort[5] from the cBio portal [http://www.cbioportal.org/] (brca_metabric.tar.gz). We manually separated clinical features into discrete and continuous features, as we did for the TCGA data. To reduce computational time and memory requirements, we focused our analysis of CNAs on the 1000 genes whose CNAs were most significantly associated to changes in the mRNA abundance of that gene—taken from Table S30 of[5]—supplemented with 124 known breast cancer driver genes[36,41,42]. We supplemented these genes with 1000 random genes not previously reported as drivers, which we used as controls. We downloaded SNVs in 173 genes from Pereira et al.[36] (somaticMutations.txt) together with the 'tumorIdMap.txt' file which maps the tumor IDs used in Pereira et al. to those used in Metabric. We used this mapping to convert the tumor IDs of Pereira et al. to Metabric IDs.

**Gene expression analysis**. For each cancer type, we focused our analyses on primary tumors (field 'sample_type' set to 'Primary Tumor' in the TCGA clinical annotation), thus removing normal control samples as well as local and distant metastases. Doing so excludes the possibility that archetypes correspond to differences in metastatic host tissues or in disease state (healthy vs cancer).

We started from a matrix of samples *times* genes (samples × genes). Entries of the matrix represent log2 normalized RPKMs (see sections "TCGA data" and "Metabric data"). The goal of the analysis was not to contrast highly expressed genes to low expressed genes in a given tumor but to identify the main changes in gene expression across tumors. To identify these changes in gene expression, we subtracted the average expression (averaging over samples) from each gene. As a result of this transformation, entries in the samples × genes matrix represented log2 fold change in expression of a given gene in a given sample compared with the average expression of that gene in that cancer type. We performed principal component analysis (PCA) on the transformed samples × genes matrix. We did not scale log2 fold changes by the standard deviation prior to PCA. As a result, large and correlated changes in gene regulation affected principal components more than small uncorrelated changes which are prone to measurement error.

**Fitting polyhedra to tumor gene expression data with ParTI**. To find polyhedra in the gene expression data from each tumor, we used the ParTI matlab software package[33]. Briefly, the input to ParTI is a large-scale dataset such as a matrix of sample × gene expression. ParTI determines the position of archetypes (vertices) in gene expression space. These archetypes define a polyhedron. How well a polyhedron fits the data is quantified by the ratio of the volume of the best-fitting polyhedron to the volume of the convex hull of the data (*t*-ratio)[23]. The *t*-ratio is

always larger than 1 and approaches 1 when the data fills a polyhedron (see Fig. 1d, glioma panel). ParTI then computes a one-sided *p*-value for the statistical significance of the polyhedron by re-computing the *t*-ratio on 1000 shuffles that conserve the distribution of loadings on each PC but not the correlation between the PCs[33].

To choose the number of archetypes, we attempted to fit three, four, or five archetypes to each cancer type. We chose the smallest number of archetypes that produced a statistically significant polyhedron ($p < 0.01$). We did not attempt to find six or more archetypes because of the limited number of tumor samples.

An R implementation of ParTI is also available at GitHub [https://github.com/vitkl/ParetoTI].

**Clustering archetypes from different cancer types.** We performed clustering analysis on the gene expression profiles of archetypes from the six cancer types with significant polyhedra (glioma, thyroid, breast, bladder, liver, and colon) to determine if different cancer types share similar tasks. We first determined the gene expression profile of each archetype as the average gene expression profile of tumors closest to the archetype (i.e., tumors in the first distance bin, see "Gene and clinical enrichment analysis" section).

Since different tissues can express different genes, we compared archetypes by collapsing the expression of 20,530 genes onto MSigDB pathways[37]. We defined the regulation of a given MSigDB pathway as the average regulation of the genes in that pathway. Averaging gene expression from individual genes into MSigDB pathways transforms the archetypes × genes matrix into a matrix of archetypes × pathways.

We focused our comparison of archetypes on the MSigDB pathways that are significantly upregulated in tumors close to at least one archetype in all 6 cancer types (FDR < 10%, see "Gene and clinical enrichment analysis").

One difficulty with visualizing which pathways are upregulated at different archetypes is that the expression of certain pathways varied strongly across tumors (e.g., immune pathways) whereas variations in the expression of other pathways were more moderate (e.g., peroxisome lipid metabolism). To visualize which pathways are upregulated at each archetype on a common color-scale, we scaled the expression of each pathway by its standard deviation across archetypes. To overcome the challenge of visualizing and interpreting hundreds of pathways on the same figure, we selected 38 pathways enriched in all tissue types and representative of each of the ten hallmarks of cancer[19] (see rows of Supplementary Fig. 11).

We clustered the archetypes from the different cancer types by Gaussian mixture modeling (mclust R package). The Bayesian Information Criteria suggested five mixtures. Archetypes clustered by tasks, not by cancer type: archetypes from a given cancer types are assigned to different Gaussian mixtures, each of which groups together archetypes from different cancer types. This observation suggests that the same tasks are relevant for different cancer types.

**Finding universal cancer archetypes.** To determine the cancer tasks shared by tumors from different cancer types, we performed ParTI on the gene expression profiles of all 3180 primary tumors from the six cancer types with significant polyhedra. Grouping gene expression profiles of primary tumors from the six cancer types combines two sources of variation: (1) differences in genes expression between tissues, (2) differences in gene expression between individual tumors of the same tissue.

In finding universal cancer types, differences in gene expression between tissues are of little interest. For example, it is not surprising for tumors from a fatty tissue like breast to show higher expression of lipid metabolism genes while brain tumors show higher expression of neuronal genes. Instead, we are interested in which genes individual tumors upregulate or repress compared with other tumors of the same tissue. To identify these changes, we subtract the mean expression (averaged over samples) from each gene prior to assembling the matrix of all samples × genes. As a result, the average expression of each gene is 0 within each tumor type, and thus across all tumors. Therefore, entries in the samples × genes matrix represent log2 fold change in the expression of a given gene in a given sample relative to the average expression of that gene in tumors from the corresponding cancer type.

TCGA tumors were collected using a different technology (RNAseq) and analysis pipeline than the metabric tumors (microarray). To ensure homogeneity in the gene expression data, we thus used the TCGA 1095 breast tumors instead of the 1970 metabric breast tumors.

We applied ParTI as described in the section "Fitting polyhedra to tumor gene expression data with ParTI". ParTI identified five archetypes, which define a polyhedron in four dimensions.

To visualize the position of the tumors in this 4D space, we projected tumors on the 2D faces of the polyhedron. Each face is defined by three archetypes. To project tumors on faces, we computed the two vectors connecting the first archetype to the two other archetypes. These two vectors define a linear basis for the face, which we orthogonalized using the Gram-Schmidt algorithm and normalized so each basis vector had norm 1. Multiplying the orthonormal matrix by a matrix of the 4D coordinates of all tumors and of the three archetypes defining the face yielded their projections on the face.

Finally, to exclude the possibility that tasks correspond to specific cancer types and confirm that tumors from individual cancer types are instead found close to multiple archetypes, we computed the fraction of tumors from individual cancer types found among the 10% tumors closest to each archetype (Supplementary Fig. 2C). If tumors from all cancer types were evenly represented close to all archetypes, we would expect these fractions to be 10%. We observe indeed that tumors from all six cancer types make up ~10% of multiple archetypes. This observation confirms that individual tasks are relevant to multiple cancer types.

**Matching tissue archetypes to universal cancer archetypes.** To determine whether tasks found in individual cancer types matched the five universal cancer tasks, we compared MSigDB pathways upregulated at each tissue-specific archetype with MSigDB pathways upregulated at each universal archetype.

We focused the comparison on MSigDB pathways significantly upregulated at the archetype (FDR < 10%) and with log2 fold change larger than 0.1 to discard pathways with minor regulation. For each pair of tissue-specific and universal archetype, we asked how many pathways are upregulated in both. We then tested if the number of pathways common to both archetypes was significantly higher than expected under the null hypothesis of random sampling from the union of pathways found at any archetype of that cancer type (hypergeometric test). We concluded that two archetypes were statistically similar when the *p*-value of the hypergeometric test was below 1% after Bonferroni correction. Results of this comparison are shown on Supplementary Fig. 2D. For visualization, the *p*-value of non-significant comparisons was set to 1.

The task of cancer-specific archetypes was assigned to that of the most similar universal archetype (gray dots on Supplementary Fig. 2D), provided that the similarity between the two archetypes was statistically significant.

Using this procedure, we found that each universal cancer task was assigned to at most one archetype within each cancer type. This is expected if tumors from different cancer types share the same tasks. One exception was found in thyroid cancer (THCA): both archetypes 1 and 3 matched the task of biomass and energy best.

Finally, archetype 4 of the metabric breast tumors (BRCA 4, which is associated to the HER2 subtype) matched none of the universal archetypes. One possible interpretation is that this archetype performs a breast cancer-specific task enabled by overactivation of HER2 signaling.

**Enrichment analysis of clinical features and genes.** Having inferred the number of archetypes and their position in gene expression space using ParTI, we characterized the task of each archetype. We did so using the methodology previously described[33].

Briefly, we considered the 50 tumors closest to each archetype. Since there were at least 250 primary tumors in each cancer type, 50 tumors correspond to at most 20% of tumors. In cases where 50 tumors represented less than 5% of all tumors, we selected the 5% tumors closest to the archetype. In these tumors, we then searched for overrepresented clinical features (Supplementary Data 2, Supplementary Data 3) and upregulated MSigDB pathways (Supplementary Data 1). We performed enrichment analysis in each cancer type (using cancer-specific archetypes, Fig. 1d) as well as by grouping tumors from all six cancer types in one analysis (using the universal cancer archetypes, Fig. 2a). When analyzing tumors from all six cancer types together, we also looked for upregulation of individual genes (Supplementary Data 1).

For each continuous feature (e.g., expression of MSigDB pathways and individual genes, quantitative clinical features such as age, recurrence-free survival, or tumor weight), we tested if the feature took significantly higher values in tumors closest to the archetype compared with other tumors (Mann–Whitney U test, two-sided). For discrete features (e.g., the presence of an SNV or of a CNA, and qualitative clinical features such as the gender of the patient, the pathological stage, and the molecular subtype of the tumor), we tested whether the feature was overrepresented in tumors closest to individual archetypes using the one-sided hypergeometric test.

We controlled for the false discovery rates (FDR) using the Benjamini-Hochberg procedure. For each archetype, we report all features with FDR < 10% and whose prevalence peaks in among tumors of the first distance bin of that archetype.

When analyzing tumors from all six cancer types, some clinical features are specific to a single-cancer type. For example, PAM50Calls are only defined for breast cancer since PAM50 is a molecular classification of breast tumors. Others features are relevant to multiple cancer types, such as the percentage of stromal cells in a tumor (% of stromal cells in a tumor). To distinguish features relevant to multiple cancer types from tissue-specific features (defined as features only defined in tumors of a single cancer type), tissue-specific features were tagged with the TCGA code of the cancer type in Supplementary Data 2 and 3. For example, we renamed PAM50Call to BRCA.PAM50Call.

Testing for the upregulation of MSigDB pathways close to an archetype has the caveat of circular inference, as genes from a given pathway are used both to infer the position of the archetype and determine its task through MSigDB enrichment analysis. To address this caveat, we use a leave-one-out strategy. For each enriched MSigDB pathway, we remove genes belonging to this pathway and infer the position of archetypes again using the remaining genes. We then test if this pathway is still upregulated in tumors close to the archetype. If not, we discard the pathway from the enrichment analysis.

In our analysis of 3180 tumors from all six cancer types, we applied this leave-out-one strategy to all MSigDB pathways with more than 100 genes. These pathways are more likely to influence the PCA and the position of the archetypes compared with pathways with less genes. We found that all MSigDB pathways identified in the original enrichment analysis were also enriched in the leave-one-out analysis. This observation suggests that archetypes are supported by large groups of genes belonging to diverse pathways, and that circular inference is not a concern in the present results.

**Inferring tasks**. To infer the task of each archetype, we used the same approach as in previous studies[29,33]: we examined what MSigDB pathways were maximally upregulated and what clinical features were maximally overrepresented in the 5% tumors closest to each archetype. We then used these MSigDB pathways and clinical features as clues to the task performed by the archetype. This section describes how upregulated MSigDB pathways and overrepresented clinical features support the tasks we inferred.

Cell division (archetype 3 in Supplementary Data 1–3): Tumors close to this archetype upregulate genes involved in the cell cycle (KEGG cell cycle, $p < 1e-60$). Upregulated genes are involved in different phases of the cell cycle, such as the M phase (regulation of mitotic cell cycle, $p < 1e-20$) and the S phase (reactome DNA replication, $p < 1e-55$). This suggests that cells from these tumors are not arrested at some point in the cell cycle but are instead dividing more than cells from other tumors. Increased cell division is consistent with the observed high cellularity in breast tumors close to this archetype ($p < 1e-6$).

Genes involved in extending telomeres are upregulated in tumors close to this archetype (reactome extension of telomeres, $p < 1e-40$). Telomeres are repeated hexanucleotides that protect the extremities of chromosomes. In somatic cells, telomeres are shortened at each division, thereby acting as a division counter which limits how many times a cell can divide before it becomes senescent or dies[63]. Cancer cells are thought to achieve replicative immortality by acquiring the capacity to extend their telomeres, which normal cells typically cannot. Upregulation of genes involved in telomere extension is thus consistent with the task of cell division.

Tumors close to this archetype upregulate genes involved in maintaining chromosome integrity (reactome activation of ATR in response to replication stress, $p = 1e-55$; KEGG mismatch repair, $p = 1e-35$), perhaps as a strategy to remain viable despite DNA damage that accumulate over cell divisions. This strategy appears only partially successful: the median tumor close to this archetype harbors 750 more CNAs than the rest of tumors ($p < 1e-40$).

Clinically, early tumors are overrepresented close to this archetype (pathologic stage: Stage IIA, $p < 1e-16$). Among breast tumors, triple negative tumors are overrepresented (BRCA, breast carcinoma estrogen receptor status: Negative, $p < 1e-16$; BRCA, breast carcinoma progesterone receptor status: Negative, $p < 1e-16$; BRCA.HER2 Final Status nature2012: Negative, $p < 1e-16$). Patients carrying these tumors show a 334 days longer recurrence-free survival than patients with other tumors on average (X_RFS, $p < 1e-5$).

Biomass and energy (archetype 2 in Supplementary Data 1–3): Tumors close to this archetype upregulate ribosomal proteins (KEGG ribosome, $p < 1e-70$; mitochondrial ribosome, $p = 1e-90$) and genes needed in translation (reactome peptide chain elongation, $p = 1e-77$; reactome translation, $p < 1-e73$). This suggests increased protein synthesis and biomass production.

Genes involved in the proteasome (KEGG proteasome, $p < 1e-65$) are also upregulated. Increasing proteasome activity is thought to be a strategy used by cancer cells to cope with the increased translation of low-quality proteins due to misfolding-causing mutations and aberrant splicing[64]. Aberrantly spliced mRNAs can contain premature stop codons. Such premature stop codons are detected as aberrantly spliced mRNAs, and degraded through the mechanism of non-sense mediated decay (NMD)[65]. NMD is upregulated in tumors close to archetype 2 (reactome NMD enhanced by the exon junction complex, $p = 1e-75$). These observations are consistent with the view that upregulation of the proteasome helps tumors cope with a proteotoxic crisis.

In addition, tumors close to archetype 2 upregulate genes needed in respiration (reactome formation of ATP by chemiosmotic coupling, $p < 1e-80$; reactome TCA cycle and respiratory electron transport, $p < 1e-76$) and in glycolysis (biocarta glycolysis pathway, $p < 1e-25$). This increase in energy producing pathways may serve to support protein synthesis and biomass production which accounts for the majority of ATP consumed in growing cells[66,67].

Clinically, tumors close to this archetype are found in patients at an early cancer stage (pathologic T: T2, $p < 0.001$).

Lipogenesis (archetype 1 in Supplementary Data 1–3): Genes upregulated at this archetype support the task of lipid metabolism reprogramming, an area of cancer research that is currently undergoing significant developments[68].

The main enzymes catalyzing de novo lipogenesis are upregulated in tumors close to this archetype: ACLY ($p < 1e-11$), ACACA ($p < 1e-16$) and FASN ($p < 1e-7$)[69]. Enzymes needed to synthesize glycosylphosphatidylinositols (GPIs), a type of phospholipid, are also upregulated in these tumors (KEGG glycosylphosphatidylinositol GPI anchor biosynthesis, $p < 1e-33$). De novo lipogenesis has been proposed to promote growth and survival of cancer cells in several ways. First, de novo lipogenesis can support the need of cancer cells for membranes in cell proliferation[69]. Second, in the hypoxic environment of tumors

cells, lack of oxygen blocks respiration which leads to an excess of reducing power (too much NADPH). De novo lipogenesis consumes NADPH and could thus help rebalance the redox balance[69]. Third, de novo lipogenesis produces lipids that are less sensitive to reactive oxidative species (ROS). ROS, which are produced by the mitochondrial activity that supports cell proliferation, can damage lipid membranes and thus endanger cell survival[70,71]. De novo synthesized lipids come saturated[72]. These saturated lipids are less sensitive to ROS than unsaturated lipids[71]. Subsequent desaturation requires oxygen, which is typically lacking in the tumor environment[72]. As a result, the increased levels of saturated lipids promoted by de novo lipogenesis may support cell survival.

Peroxisomes are organelles most known for clearing ROS. But they also carry out other functions, such as synthesizing lipids (in particular ether lipids) and shortening long fatty acid chains for use in mitochondrial metabolism[73,74]. Tumors close to archetype 1 upregulate peroxisomal genes (peroxisomal part, $p = 1e-40$) and genes involved in peroxisomal lipid metabolism ($p < 1e-22$). Peroxisomal genes upregulated in tumors close to archetype 1 include ABCD3 ($p = 1e-22$) which transports fatty acids to peroxisomes, as well as ACOX1 ($p = 1e-9$), ACOX3 ($p < 1e-5$), and AMACR ($1e-17$) which carry out beta-oxidation of long fatty acids in peroxisomes[74]. Although the exact function of peroxisomes in cancer is not clearly understood yet, beta-oxidation of long fatty acid chains in peroxisomes could represent an alternative energy source to glucose[75].

Clinically, tumors close to this archetype tend to be early stage (pathologic stage: Stage IIA, $p < 0.001$) and well differentiated (neoplasm histologic grade: Low Grade, $p < 0.001$). The median patient carrying these tumors was 8.5 years older than the median patient carrying other tumors ($p < 1e-4$). Breast cancer tumors close to this archetype are enriched with hormonal cancer (BRCA.ER Status nature2012: Positive, $p < 1e-9$, BRCA.PR Status nature2012: Positive, $p < 1e-8$).

Immune interaction (archetype 4 in Supplementary Data 1–3): Tumors close to this archetype upregulate genes expressed in immune cells (KEGG allograft rejection, biocarta tcytotoxic pathway) and related to inflammation (biocarta inflam pathway, reactome interferon gamma signaling). There is also upregulation of the PD-1 and CTLA-4 pathways which inhibit immune response (reactome PD-1 signaling, $p < 1e-55$; BIOCARTA CTLA-4 PATHWAY, $p < 1e-58$). This suggests an archetype characterized by the invasion of tumors by immune cells, but whose action is inhibited. Consistent with the invasion of the tumors by immune cells, tumors close to this archetype show loss of tissue identity and poor differentiation in histological examinations (neoplasm histologic grade: High Grade, $p = 0.006$).

The median tumor close to this archetype has five more SNVs than other tumors (number of SNVs, $p = 0.0002$). This association of the number of SNVs to immune invasion is consistent with previous reports that PD-1 blockage therapy is most effective against tumors with higher mutational burden[76].

Clinically, patients with these tumors are at a more advanced disease stage (pathologic stage: Stage III, $p < 1e-5$) than archetypes 1–3 (Stage II). This difference suggests that the task of immune interaction becomes relevant to tumors at a later stage than the tasks of cell division, biomass and energy, and lipogenesis.

Invasion and tissue remodeling (archetype 5 in Supplementary Data 1–3) : Tumors closest to this archetype upregulate genes involved in the remodeling of the extracellular matrix (extracellular matrix structural constituent, $p < 1e-65$; extracellular structure organization and biogenesis, $p < 1e-65$; reactome degradation of the extracellular matrix, $p < 1e-40$). Disorganization of the ECM participates in tumor progression and metastasis[77]. The second most upregulated MSigDB pathways in these tumors is collagen ($p = 1e-57$), which suggests the presence of cancer associated fibroblasts (CAFs). Accordingly, the average percentage of stromal cells in tumors close to this archetype was 11 points higher compared with other tumors ($p = 0.0006$).

Tumors close to this archetype upregulate genes involved in invasion and metastases such as cell migration ($p < 1e-80$), and angiogenesis (regulation of angiogenesis, $p < 1e-64$). Consistent with the task of invasion, tumors close to this archetype invaded 1.8 more lymph nodes than other tumors on average (number of lymph nodes positive by H&E, $p < 0.0005$). Lymphatic and venous invasion is overrepresented in colon tumors close to this archetype ($p < 0.0004$), as well as lymphovascular invasion in bladder tumors (BLCA.lymphovascular invasion present: YES, $p = 0.003$). Patients carrying tumors close to this archetype are at the most advanced disease stage of all five archetypes (pathologic stage: Stage IIIC, $p < 1e-7$). Thus, both gene expression and clinical features support the task of invasion.

In addition, tumors close to this archetype activate signaling pathways involved in development and tissue repair/homeostasis: Hedgehog, Insulin-like growth factor, fibroblast growth factor, Wnt, TGFβ, Ras (PID HEDGEHOG 2PATHWAY, $p < 1e-80$; reactome regulation of insulin-like growth factor IGF, $p < 1e-65$; reactome FGFR ligand binding and activation, $p < 1e-77$; PID WNT signaling pathway, $p < 1e-61$; KEGG TGF beta signaling pathway, $p < 1e-75$; RAS GTPASE activator activity, $p = 1e-72$). Tumors are thought to hijack these programs to support cancer progression[19]. In the case of TGFβ signaling for example, epithelial tumors hijack a program normally used in development and wound healing in which epithelial cells acquire the ability to invade into a tissue and resist apoptosis[78].

These tumors also upregulate immune pathways, although at lower levels than tumors close to the immune interaction archetype. In addition, the identity of upregulated immune pathways differs between archetypes 4 and 5. For example,

tumors close to the invasion and tissue remodeling archetype upregulate the lectin pathway (biocarta lectin pathway is the fourth most upregulated MSigDB gene group, $p < 1e-59$). In contrast, the lectin pathway is not upregulated among tumors close to the immune interaction archetype.

Finally, tumors close to this archetype upregulate neural pathways (regulation of neurogenesis, voltage-gated sodium channel activity). Voltage-gated sodium channel genes define the single most upregulated MSigDB pathway among tumors close to this archetype. Upregulation of these pathways could result from cancer cells hijacking pathways normally active during neuron migration[19]. It could also be due to perineural invasion, a process in which cancer cells spread along nerves[79]. Alternatively, this upregulation of neural pathways could be the sign of neural stimulation or modulation of the tumor microenvironment[80].

**Matching cancer tasks to cancer hallmarks**. To compare the five universal cancer tasks to the ten cancer hallmarks defined by[19], we picked 1–3 MSigDB pathways representative of each cancer hallmark (Fig. 2b). We then plotted the $p$-value quantifying the statistical significance of the upregulation of each MSigDB pathway in the 5% tumors closest to each archetype, as described in the "Enrichment analysis" section.

**Drug sensitivity analysis in breast cancer cell lines**. To test if cancer cells that specialize in a task are more sensitive to drugs that interfere with the task, we used the gene expression and drug sensitivity data in 56 breast cancer cell lines of ref. [14].

We determined the position of the 56 cancer cell lines relative to the four breast cancer archetypes. To do so, we only considered the 14,844 genes whose expression was quantified in both the 1970 Metabric breast tumors[5] and in the 56 breast cancer cells lines[14]. We then performed quantile normalization on the joint samples × genes matrix of log2 gene expression using Bioconductor's normalize. quantile function[81]. To focus the analysis on the diversity in gene expression within cancer cell lines, we subtracted the mean gene expression profile out of each cell line. As a result, each gene had an average expression of zero, so that expression of a given gene in a certain cell line represented log2 fold changes relative to the mean expression across all breast cancer cell lines. We then projected both cell lines and breast cancer archetypes on the first three principal components of breast tumors, which were determined while finding the archetypes of breast cancer (see section "Fitting polyhedra to tumor gene expression data with ParTI"). In computing the projection, we kept only the 14,844 dimensions corresponding to genes quantified both in tumors and cell lines. For visual reference, the 1970 Metabric tumors were also projected onto the same space.

GR AOC is a measure of drug sensitivity robust to cell-to-cell variations in growth rates[38]. We determined how the GR AOC varied as a function of euclidean distance to the archetype by grouping cell lines in four distance bins. For each bin $i$, we computed the median GR AOC $G_i$ and the within-bin standard error $\sigma_i$. Because some drugs were assayed against only some of the 56 cell lines, we discarded drugs with <5 cell lines per bin.

For each archetype, we looked for drugs whose potency peaked in cell lines closest to the archetype, and then monotonically decreases away from it. To address the concern that the small sample size (56 breast cancer cell lines vs 1970 breast tumors per cancer type) can produce false positive drug-archetype pairs, we used a more stringent criterion to identify drugs effective against individual archetypes than the criteria used to find overrepresented MSigDB pathway and clinical features (see "enrichment analysis" section). We scored each drug by the product of the decrease in GR AOC in consecutive bins plus twice the standard error within bin, $\Pi_i$ ($G_i - G_{i+1} + 2 <\sigma_i>$), with $<\sigma_i>$ the median standard error across all bins. Drugs which score high on this scheme see their GR AOC peak close to the archetype and then steadily decrease away from it. If GR AOC increased by more than twice the standard error in any consecutive bins, the score was set to 0. This tolerates small bin-to-bin increase in GR AOC which could be due to measurement error or cell-to-cell variability. Finally, we tested if GR AOC was significantly higher in cell lines of the first distance bin compared with all other bins (FDR < 10%, Mann–Whitney test, see "enrichment analysis" section). In doing so, everolimus, rapaycin, and temsirolimus were grouped together as they share the same mechanism (mTOR inhibition, Fig. 2g).

Of the top ten-scoring drugs, six appear on Fig. 2e–h. In addition to these six drugs, Triciribine which inhibits the Akt kinase (an mTOR activator) is effective against the biomass and energy archetype ($p = 0.01$), consistent with the sensitivity of the biomass and energy archetype to mTOR inhibitors (Fig. 2g). Gemcitabine, another drug in the top 10, was not statistically significant at FDR < 10% ($p = 0.11$). The identity of the remaining drugs within the top 10 was kept confidential by the authors of the dataset[14], thereby preventing further interpretation.

**Obtaining driver and passenger alterations**. To test the hypothesis that driver SNVs and CNAs are knobs that tune tumor gene expression towards specific tasks, we first compiled lists of (1) drivers alterations, (2) passengers alterations, and (3) alterations commonly found in cancer although not thought to be drivers in the cancer type of interest. Second, we quantified whether these alterations pushed tumor gene expression along the cancer front or away from it.

For each cancer type except breast cancer, we obtained lists of driver genes from the IntOGen database[40]. We focused on known driver genes, for which there is

causal evidence of their implication in cancer according to the Cancer Gene Census[82]. SNVs and CNAs in these known driver genes were labeled driver alterations.

In breast cancer, instead of using the IntOGen database, we took advantage of extensive efforts in classifying driver SNVs and CNAs as oncogenes and tumor suppressors[36,41,42]. We defined driver alterations from: (1) genes lying within minimal amplification regions, upregulated as a result of this amplification, and with significant experimental evidence of their involvement in cancer development by Santarius et al. (class I, II, and III)[42]. We defined these genes as oncogenes. (2) genes with a high proportion of recurrent SNVs (for oncogenes) or a high proportion of inactivating mutations (for tumor suppressors) in the survey of 2433 primary breast tumors of Pereira et al.[36]. (3) genes commonly amplified, or carrying mis-sense or inframe mutations (for oncogenes), and genes commonly deleted or carrying frameshifting or non-sense mutations (for tumor suppressors) in a survey of 560 breast tumors[41].

SNVs corresponding to genes identified as oncogenes or tumor suppressors in any of these three studies were defined as driver SNVs. Amplifications (weak or strong) corresponding to genes identified as oncogenes were defined as driver CNAs. Deletions (weak or strong) corresponding to genes identified as tumor suppressors were also defined as driver CNAs.

Individuals SNVs and CNAs that did not concern known driver genes were called other cancer genes if they appeared in the list of genes frequently amplified or mutated in cancer, listed in Table S2 of ref. [6]. Finally, SNVs and CNAs in genes not found in the two lists were called passengers.

**Assessing alignment of SNVs and CNAs to front**. We represented the average effect of each alteration on gene expression as a vector, connecting the centroid of tumors having the alteration to the centroid of tumors without the alteration (Fig. 3a). To quantify whether an alteration pushes gene expression along the cancer front or away from it, we computed the angle between the alteration vector and the cancer front. The cancer front was defined as the linear subspace that contains the archetypes. The angle between the cancer front and the alteration vector was computed by projecting the alteration vector on the cancer front, and by computing the scalar product between the alteration vector $u$ and its projection $v$. The scalar product $<u, v>$ was converted to an angle $\alpha$ (in degrees) using the inverse cosine, $\alpha = 180 \cos^{-1}(<u, v>/|u|.|v|)/\pi$.

To quantify the statistical significance of the alignment of an alteration vector to the cancer front, we randomly assigned alterations to tumors and computed the corresponding alteration vectors. By repeating this procedure, we obtained $10^5$ shuffled control vectors and their angles to the cancer front. We used the distribution of angles of shuffled controls—which was independent of the frequency of the alteration—to estimate the $p$-value that individual alteration vectors are aligned to the cancer front. In principle, the (one-sided) $p$-value can be estimated from the fraction of shuffled controls with a smaller angle to the cancer front than a given alteration. An issue with this approach is that small angles are rare among shuffle controls. Thus, the $p$-values of strongly aligned alterations is imprecise. Precise estimation of the $p$-value of strongly aligned alterations is important when correcting for multiple testing.

To better estimate the $p$-values, we first computed the empirical cumulative distribution of the angle distribution of shuffle controls $F(\alpha)$. $1-F(\alpha)$ is the $p$-value of an alteration of angle $\alpha$. Because the function $\log[1-F(\alpha)]$ is much flatter than $F(\alpha)$, we focused on approximating $\log[1-F(\alpha)]$. $\log[1-F(\alpha)]$ was fitted to a fifth order polynom. In fitting the polynom, we used only angles $\alpha$ supported by at least ten shuffled controls. By extrapolating the polynom to small angles $\alpha$, we computed the $p$-value for each alteration vector to be aligned to the cancer front. Finally, we controlled the FDR using the Benjamini-Hochberg procedure[83].

**Comparing intra-tumor heterogeneity and inter-tumor diversity**. To characterize associations between clonal heterogeneity and cancer tasks, we colored metabric tumors according to their mutant allele tumor heterogeneity (MATH) score[36,84] (Supplementary Fig. 4A). MATH scores were communicated by Oscar Rueda, Caldas lab, CR UK. We tested for enrichment of the MATH score among tumors closest to the different archetypes and among tumors closest to center of mass of the gene expression data using the ParTI methodology.

To compare intratumor heterogeneity and intertumor diversity, we first analyzed the single-cell gene expression using the Seurat package[85]. Genes present in <5% of cells and outlier cells with <1500 or more than 8000 detectable genes were excluded from the analysis. Gene expression data was normalized and scaled by the RNA count of each cell, the fraction of mitochondrial RNA, and the tumor of origin, as was done in the original study[47]. Hematopoetic cells were excluded based on the expression of the PTPRC marker. CAFs and endothelial cells also identified based on other markers (PDGFRA, ZEB1, and ACTA2 for CAFs, PECAM1 for endothelial cells). After this filtering, 650 single-cancer cells from six breast tumors were left for analysis.

We projected the single-cancer cells on the first three principal components of metabric tumors[5]. To do so, we focused on 1964 genes expressed in at least 50% of single-cancer cells and profiled in the metabric tumors, so that gene expression could be quantified in most cells, allowing projection of these cells on the space defined by the first three principal components of the metabric tumors.

To assess if intratumor heterogeneity and intertumor diversity are oriented in common directions of gene expression space, we computed the fraction of variance in gene expression explained by principal components computed on gene expression of metabric tumors. This fraction is necessarily bounded by the fraction of variance explained by principal components computed on gene expression from single cells. In other words, if intratumor heterogeneity and intertumor diversity were perfectly aligned in gene expression space, the fraction of variance explained by metabric principal components would equal the fraction of variance explained by single-cell principal components. We used five PCs because of an elbow in the fraction of variance explained by single-cell PCs curve (Supplementary Fig. 4B). The first five metabric PCs explain $25.4 \pm 3\%$ of the variance explained by the first five single-cell PCs. This percentage is robust to using a different number of PCs (Supplementary Fig. 4C): varying the number of PCs from 5 to 50 keeps the percentage in the 22–27% range. To test if the fraction of variance explained by metabric PCs can be attributed to random structures in gene expression data, we shuffled the metabric gene expression by resampling expression of each gene. Doing so preserved the variance in the expression of each gene but destroyed correlations in the expression of different genes. The first five PCs of this shuffled dataset explain $1.5 \pm 0.4\%$ of the variance explained by single-cell PCs. A $t$-test supports the hypothesis that the first five metabric PCs explain significantly more variance in single cell gene expression than shuffled metabric PCs ($p = 0.0004$, two-sided). Thus, intertumor diversity explains a significant fraction of intratumor heterogeneity.

**Reporting summary**. Further information on research design is available in the Nature Research Reporting Summary linked to this article.

## Data availability

Source data files to reproduce the analyses can be downloaded at: https://doi.org/10.17632/2r9h9xzwm3.1

The TCGA data used in this study was downloaded from RSEM-normalized HiSeqV2 gene expression data (log2 RPKMs) from the TCGA data portal. The TCGA data portal is now retired but the data can be retrieved from the Genomics Data Commons portal of the National Cancer Institute [https://portal.gdc.cancer.gov/]. The starting point of our analyses were the 'genomicMatrix' files which contain expression levels for 20,530 human genes in 15 cancer types. We considered all cancer types with at least 250 primary tumor samples: BLCA, BRCA, CESC, COAD, HNSC, KIRC, LGG, LIHC, LUAD, LUSC, OV, PRAD, STAD, THCA, and UCEC. The data for the 1970 tumors of the METABRIC data were downloaded from the cBio portal [http://www.cbioportal.org/study/summary?id=brca_metabric]. We describe in detail how we obtained and processed the data in the "Methods" section.

## Code availability

The code to reproduce the analyses can be downloaded at https://doi.org/10.17632/2r9h9xzwm3.1. Analyses were performed in R 3.4 and Matlab R2016b.

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

## Acknowledgements

The authors would like to thank Ruthie Shouval, Ravid Straussman, and members of the Alon lab for discussions. We thank Oscar Rueda for discussions and assistance in obtaining data. This work was supported by the Minerva foundation. U.A. is the incumbent of the Abisch-Frenkel Professorial Chair and acknowledges funding by BSF-NSF-NIH-CRCNS. Research in C.C. laboratory is funded by Cancer Research UK and an European Research Council Advanced Grant. C.C. was supported by a Weston Visiting Professorship at the Weizmann Institute. J.H. acknowledges the support of the Swiss National Science Foundation (P300P3-158472) and the Swiss Society of Friends of the Weizmann Institute.

## Author contributions

Conceptualization: J.H., C.C. and U.A.; Methodology: J.H., P.S., C.C. and U.A.; Software, formal analysis and investigation: J.H., P.S., N.B., A.Z. and H.S.; Writing: J.H., C.C. and U.A.; Supervision: J.H. and U.A.; Funding Acquisition: J.H., C.C. and U.A.

## Competing interests

C.C. is a member of AstraZeneca's External Science Panel and is a recipient of research grants (administered by the University of Cambridge) from AstraZeneca, Genentech, Roche, and Servier. The remaining authors declare no competing interests.
