## [Peer Review File · Nature Communications]

Reviewers' comments:

Reviewer #1 (Remarks to the Author):

This is a fascinating paper that provides the first application of the analysis of molecular and phenotypic functional tradeoffs to the evolution of cancer cells, in any sort of comprehensive and empirical way. The study is highly innovative in its integration of evolutionary-ecological theory regarding tradeoffs to cancer biology, and its use of the method of Pareto task inference in this context. The conclusions are notably original in showing evidence of multiway tradeoffs in major forms of human cancer, which has important general implications for how cancer is understood and how therapies can be applied. The results should be of interest to specialists in cancer biology and genetics (who will gain new theoretical as well as empirical perspectives), to researchers in evolutionary biology and genetics, and to biologists more broadly.

A number of points should, however, be addressed to help improve the article:

(1) The variation among forms of cancer in the results relevant to tradeoffs is quite striking, but it is not addressed even to the point of raising plausible hypotheses. Might it be related to variation in the extent of de-differentiation of the cancer cells, or to their original cellular sources (eg stem cells, or other)? Ages of onset? Numbers of mutations required for conversion to cancer? It is quite curious that tradeoffs are so evident for some cancers but not others.

(2) The authors discuss 'needs' at several points in the article; however there are no needs per se in the evolutionary process, just natural selection and other population-genetic processes. Rewording is required.

(3) There are more than five (there are six) 'hallmarks' of cancer as described in Hanahan and Weinberg; why are only five evident here?

(4) Paragraph starting 'To test whether...'. The authors should add a few sentences that provide a simpler and more-intuitive explanation of the Pareto method, and how statistical significance is tested here.

(5) By 'archetypes' do the authors (also) mean 'specialists'?

(6) How does the considerable genetic diversity *within* tumors, that is ultimately responsible for most mortality, fit with the specialist/tradeoffs framework? Might different subclones of tumors specialize for different tasks (depending in part for example on position in the tumor)? (i.e., cooperation among cells within tumors, as described originally by Axelrod)?

(7) How does the time-trajectory of cancers (the order of acquisition of the hallmarks/functions) fit with the tradeoff framework?

(8) The authors should note that the presence and strength of tradeoffs commonly depends on the amount of resources available (ie more resources, less tradeoff). This point is well-studied in evolutionary ecology and should be acknowledged here, and its potential application to cancer should be noted. At the end of the first Discussion paragraph, the authors say that tradeoffs give negative correlations; this need not be true if absolute resources vary to a high extent.

(9) Discussion paragraph starting 'We considered here total...'. Evolution along lines of genetic least resistance refers to patterns of genetic correlation and heritability that define evolutionary trajectories for multiple traits, within (and across) species. The parallel/metaphor with regard to cancer is not made sufficiently clear by the authors.

Dr. Bernard Crespi

Reviewer #2 (Remarks to the Author):

The manuscript submitted by Alon and colleagues takes an innovative new approach to uncovering molecular task optimization as a route to explaining inter-patient variation across multiple cancer types. To my knowledge this is the first exposition of multi-task evolutionary theory to formalize how tumors navigate task tradeoff as they acquire specialized phenotypes. The authors implement Pareto task inference to analyze the TCGA, Metabric and other datasets. The major claims are as follows: (i) they identify tradeoffs between five universal tasks shared across cancer types, (ii) show that tumors that specialize in a task are differentially sensitive to drugs that disrupt that task, and (iii) demonstrate that each driver mutation moves gene expression towards specific archetypes and hence towards specialization in specific tasks. Overall, this is an interesting and novel approach that has potential to function as a formalism toward drug sensitivity prediction.

Critique:

1. In general, the biggest issue this reviewer sees is the lack of consideration of confounding factors in the authors' analysis. These include, for example well-defined molecular subtypes and within tumor heterogeneity (comments below).
2. The major claim made is that tumors specializing in specific tasks are found closer to the vertices of the simplex. Unfortunately this is not intuitively or convincingly presented. Figures 2&3 do not seem to have any points near the vertices. Also in most cases the density of the distribution (in particular Fig 2) is not evident due to the manner in which the data is plotted. This needs addressing to convince the readership of the central claims of the paper.
3. Pg 4. There may be multiple gene expression routes to the archetypes. How does the method handle this? Would the signal be diluted as the number of routes increases? Could this explain the 'clouds' in expression space?
4. Pg 4. How much does tumor purity dictate how sharply the tumor can be resolved as archetypes?
5. Pg 5. What is the relationship between clonal diversity at the mutational level and the ability to resolve sharply defined archetypes?
6. Pg 5. how do the archetypes map to known cancer subtypes (iclusters in Metabric or PAM50 in TCGA breast cancer)?
7. Pg 6. 2nd last paragraph. Available single cell data from various cancer types could resolve this convincingly.
8. Pg 8. "Mutations in TP53 in breast cancer...". This is association and not causation. There could easily be confounding effects that yield the association such as known subtypes enriched for specific driver mutations. Thus the association may relate to a cell type of origin better able to tolerate TP53 mutation.
9. That the cell lines specializing in specific tasks are more sensitive to the drugs that inhibit these tasks is not particularly surprising? It is not clear how this particular analysis is better able to resolve this than traditional pathway analysis. Could the authors elaborate?
10. The paper lacks any benchmarking data. What are the appropriate alternative approaches to this type of inference? As the paper has a methodological component, a comparative component to complementary methods should be included.
11. Is the source code for the model and analysis available in a public repository (e.g. github?)

Reviewer #1

This is a fascinating paper that provides the first application of the analysis of molecular and phenotypic functional tradeoffs to the evolution of cancer cells, in any sort of comprehensive and empirical way. The study is highly innovative in its integration of evolutionary-ecological theory regarding tradeoffs to cancer biology, and its use of the method of Pareto task inference in this context. The conclusions are notably original in showing evidence of multiway tradeoffs in major forms of human cancer, which has important general implications for how cancer is understood and how therapies can be applied. The results should be of interest to specialists in cancer biology and genetics (who will gain new theoretical as well as empirical perspectives), to researchers in evolutionary biology and genetics, and to biologists more broadly.

We thank the reviewer for this endorsement.

A number of points should, however, be addressed to help improve the article:

1. The variation among forms of cancer in the results relevant to tradeoffs is quite striking, but it is not addressed even to the point of raising plausible hypotheses. Might it be related to variation in the extent of de-differentiation of the cancer cells, or to their original cellular sources (eg stem cells, or other)? Ages of onset? Numbers of mutations required for conversion to cancer? It is quite curious that tradeoffs are so evident for some cancers but not others.

In the revised discussion, we now raise hypotheses about what could explain why different cancer types face different trade-offs: "One question raised by our findings is why tumors vary in their degree of specialization in the different tasks. Some of this variation is associated to tumor stage: early-stage tumors are found closer the cell division, biomass&energy and lipogenesis archetypes while late tumors are more often found close to the immune interaction and invasion&tissue remodeling archetypes. This suggests that variations in the degree of specialization of tumors in different archetypes could stem from changes in the nature selection trade-offs during tumor progression, with proliferation being more important during early stages and survival during later stages. Other variables such as patient age, driving genetic alteration, type of genetic damages (copy number alterations vs single nucleotide variations) are also associated to different archetypes, suggesting that these variables could also determine why different tumors specialize in different archetypes (Table 1). Interestingly, different sets of trade-offs are identified in different cancer types (Fig. 2B). This suggests that factors such as the cell of origin or differences in the micro-environment of different host tissues could explain why different tumors specialize in different tasks. Untangling the causalities that underlie these variations could be the aim of future research."

2. The authors discuss 'needs' at several points in the article; however there are no needs per se in the evolutionary process, just natural selection and other population-genetic processes. Rewording is required.

We have now replaced 'needs' with wordings that are in line with evolutionary concepts. In the introduction, page 3 now reads "There is selection on cells that grow in a challenging environment to express survival genes, which comes at the expense of growth genes (Scott *et al.*, 2010; Shoal *et al.*, 2012)." On the same page, we now write: "Furthermore, cancer cell growth and survival is conditioned on fulfilling multiple tasks, including growth, stress resistance, interaction with the immune system and so forth (Hanahan and Weinberg, 2011). Each task requires a different profile of gene expression – ribosomes for growth and stress proteins for survival."

3. *There are more than five (there are six) 'hallmarks' of cancer as described in Hanahan and Weinberg; why are only five evident here?*

We now comment on this point in the discussion: “The 5 tasks match the 10 hallmarks described by Hanahan & Weinberg (Hanahan and Weinberg, 2011) suggested. Our data-driven approach identifies 5 tasks, so that each task combines several hallmarks. One explanation for this is that implementing a task can require several hallmarks. For example, the hallmarks of evasion of growth suppressors and enabling replicative immortality both support the cell division archetype. The archetype of immune interaction combines the hallmarks of resisting cell death, avoiding immune destruction and tumor-promoting inflammation (Fig. 2C).”

4. *Paragraph starting 'To test whether...'. The authors should add a few sentences that provide a simpler and more-intuitive explanation of the Pareto method, and how statistical significance is tested here.*

In the results section, we now write:

“It thus makes sense to apply Pareto optimality theory to the tumor situation. The theory predicts that such trade-offs lead to a characteristic pattern: gene expression, averaged over all of the cell types in the tumor, is arranged within a polyhedron in gene expression space, a geometric structure with flat sides and sharp vertices (Shoval *et al.*, 2012). For example, a polyhedron with 2 vertices is a line (Fig. 1A). A polyhedron with 3 vertices is a triangle (Fig. 1B). 4 vertices describe a tetrahedron (Fig. 1C). The vertices of the polyhedron, called archetypes, are profiles optimal for one of the tasks (Fig. 1A). Specialist tumors at a task lie close to a vertex, and generalists lie in the middle of the polyhedron (Fig. 1B). Tumors outside the polyhedron are sub-optimal, and will not be selected.”

In the next paragraph, we now also state how statistical significance is tested:

“We used the Pareto Task Inference (ParTI) software package (Hart *et al.*, 2015) which fits lines, triangles, tetrahedra and so on to data, finds the best fit polyhedron. The statistical significance of fitting a polyhedron to the data is assessed by the t-ratio test (Shoval *et al.*, 2012; Hart *et al.*, 2015) which compares how well the data fills the polyhedron compared to randomized datasets (inset of Fig. 1D).”

5. *By 'archetypes' do the authors (also) mean 'specialists'?*

An archetype is indeed the gene expression profile optimal for a certain task. Thus, tumors located closed to an archetype in gene expression space are specialists at that task.

We now clarify this point on page 4: “The vertices of the polyhedron, called archetypes, are profiles optimal for one of the tasks (Fig. 1A). Specialist tumors at a task lie close to a vertex, and generalists lie in the middle of the polyhedron (Fig. 1B).”

On page 7, “tumors near an archetype in gene expression space are task specialists”.

6. *How does the considerable genetic diversity *within* tumors, that is ultimately responsible for most mortality, fit with the specialist/tradeoffs framework? Might different subclones of tumors specialize for different tasks (depending in part for example on position in the tumor)? (i.e., cooperation among cells within tumors, as described originally by Axelrod)?*

We now address how genetic diversity within tumors fits within the specialist/generalist framework by adding 2 figures (Fig. 4, Fig. S4). We find that individual clones can be task specialists but also generalists and that intra-tumor heterogeneity is partly explained by inter-tumor diversity. In the discussion, we now write:

“We considered so far diversity in gene-expression between different tumors. We now ask about how this inter-tumor variation relates to the variation between single cancer cells inside a tumor (Tirosch *et al.*, 2016; Kim *et al.*, 2018). One possibility is that each tumor is composed of a combination of specialist cells, that specialize in the different cancer tasks. Each specialist cell thus has an expression program at one of the archetypes. The position of the entire tumor in the polytope is determined by the ratios of these specialist cell types.

This possibility is inconsistent with the data: if tumors were made of specialist cells, specialist tumors should be nearly most homogeneous in a single type of specialist cells, whereas generalist tumors should be heterogeneous. We tested this by considering heterogeneity in each tumor according to a mutation-based score. We find that specialist breast tumors can be either homogeneous or heterogeneous (Fig. S4A), and that tumors that specialize in the cell division task are more heterogeneous than generalist tumors ($p=1.6 \times 10^{-5}$, Fig. S4A).

We therefore asked whether single cancer cells within a tumor span a continuum of gene expression. Projecting single cancer cell gene-expression data onto the space defined by the first three principal components of breast tumor gene expression confirms that single cells from individual tumors can be specialist or generalists at the different tasks (Fig. 4A).

Finally, we used a concept from ecology (Schluter, 1996; Sheftel *et al.*, 2018) to hypothesize that the variation between individual cells in a tumor will fall along the subspace defined by variation between different tumors. The reason for this is that the evolution of cells within a given tumor occurs under tradeoffs between the same tasks that apply to all tumors (Shoval *et al.*, 2012). The ecological analogy, called “evolution along lines of least genetic resistance”, is that variation within given species in traits falls along the directions defined by variation between species.

To test this, we analyzed single-cell transcriptomic data from breast tumors (Karaayvaz *et al.*, 2018). We compared the fraction of the variance in single cancer cell gene expression explained by the first 5 principal components (PCs) of the single cell data to the variance in single cancer cell gene expression explained by the first 5 PCs of Metabric breast tumors (Curtis *et al.*, 2012).

If Metabric tumors and single cancer cells vary along the same directions in gene expression space, PCs computed on Metabric tumors will explain as much variation in single cancer cell gene expression as PCs computed on the single cell data. On the other hand, if single cell gene expression varies along directions different from Metabric tumors, PCs computed on the Metabric data will explain a negligible fraction of the variation in single cell gene expression because the space of gene-expression has thousands of dimensions along which tumors and cells could vary.

We find that the first 5 Metabric tumor PCs explain 25.4% of the variation explained by the first 5 single cancer cells PCs, with little change if different numbers of PCs are used between 5 to 50 (Fig. 4, Fig. S4B-C). This alignment between inter-tumor and intra-tumor variation is much higher than expected by chance: the same number of random directions in gene expression space only explains 1.5% of intra-tumor variation ($p < 0.001$, Fig. 4). The remaining ~75% of the variation may be due to stochasticity in single-cell gene expression, measurement error and neutral variation.”

7. How does the time-trajectory of cancers (the order of acquisition of the hallmarks/functions) fit with the tradeoff framework?

This is difficult question to answer because: 1. collecting time-course of tumor gene expression is difficult in clinical practice, 2. to make data comparable and limit technical artifacts, time-

course data would need to be gathered using the same technical platform as TCGA or Metabric. We are not aware of such data.

However, we can suggest an answer by analyzing how the importance of different tasks changes as a function of tumor stage. In the discussion, we now write:

“Early-stage tumors are found closer the cell division, biomass&energy and lipogenesis archetypes while late tumors are more often found close to the immune interaction and invasion&tissue remodeling archetypes. This suggests that variations in the degree of specialization of tumors in different archetypes could stem from changes in the nature selection trade-offs during tumor progression, with proliferation being more important during early stages and survival during later stages.”

8. *The authors should note that the presence and strength of tradeoffs commonly depends on the amount of resources available (ie more resources, less tradeoff). This point is well-studied in evolutionary ecology and should be acknowledged here, and its potential application to cancer should be noted. At the end of the first Discussion paragraph, the authors say that tradeoffs give negative correlations; this need not be true if absolute resources vary to a high extent.*

We now refer to evolutionary research on how trade-offs come about when resources are limited, and how recognizing this is relevant to cancer: “Trade-offs are often found when resources – nutrients, time, space – are limited and have been well-studied in evolutionary ecology (Futuyma and Moreno, 1988). A well-known example of trade-offs is found in bacteria: cells that grow faster are more sensitive to stress and antibiotics (Balaban *et al.*, 2004). There is selection on cells that grow in a challenging environment to express survival genes, which comes at the expense of growth genes (Scott *et al.*, 2010; Shoval *et al.*, 2012). A similar trade-off is found in cancer cells (Aktipis *et al.*, 2013). For example, cancer cells exposed to hypoxia can survive by invading the tissue surrounding the tumor (Sullivan and Graham, 2007; Hatzikirou *et al.*, 2012) which can come at the cost of a reduced proliferative activity (Evdokimova *et al.*, 2009; Tsai and Yang, 2013).”

We now also note that negative correlations are only expected if resources are consistently limiting: page 9 now reads “If trade-offs are at play, and resource limitation consistently enforce such trade-offs, one should find negative correlation between the different measures of performance.”

9. *Discussion paragraph starting 'We considered here total...'. Evolution along lines of genetic least resistance refers to patterns of genetic correlation and heritability that define evolutionary trajectories for multiple traits, within (and across) species. The parallel/metaphor with regard to cancer is not made sufficiently clear by the authors. Dr. Bernard Crespi*

We now clarify this point: “Finally, we used a concept from ecology (Schluter, 1996; Sheftel *et al.*, 2018) to hypothesize that the variation between individual cells within a tumor will fall along the subspace defined by variation between different tumors. The reason for this is that the evolution of cells within a given tumor occurs under tradeoffs between the same tasks that apply to all tumors (Shoval *et al.*, 2012). The ecological analogy, called “evolution along lines of least genetic resistance”, is that variation in traits within given species falls along the directions defined by variation between species.”

Reviewer #2

The manuscript submitted by Alon and colleagues takes an innovative new approach to uncovering molecular task optimization as a route to explaining inter-patient variation across multiple cancer types. To my knowledge this is the first exposition of multi-task evolutionary theory to formalize how tumors navigate task tradeoff as they acquire specialized phenotypes. The authors implement Pareto task inference to analyze the TCGA, Metabric and other datasets. The major claims are as follows: (i) they identify tradeoffs between five universal tasks shared across cancer types, (ii) show that tumors that specialize in a task are differentially sensitive to drugs that disrupt that task, and (iii) demonstrate that each driver mutation moves gene expression towards specific archetypes and hence towards specialization in specific tasks. Overall, this is an interesting and novel approach that has potential to function as a formalism toward drug sensitivity prediction.

We thank the reviewer for this endorsement.

Critique:

1. In general, the biggest issue this reviewer sees is the lack of consideration of confounding factors in the authors' analysis. These include, for example well-defined molecular subtypes and within tumor heterogeneity (comments below).

We thank the reviewer for inviting us to interpret our findings in view of known cancer subtypes and tumor heterogeneity (see our response to specific comments below).

2. The major claim made is that tumors specializing in specific tasks are found closer to the vertices of the simplex. Unfortunately this is not intuitively or convincingly presented. Figures 2&3 do not seem to have any points near the vertices. Also in most cases the density of the distribution (in particular Fig 2) is not evident due to the manner in which the data is plotted. This needs addressing to convince the readership of the central claims of the paper.

We thank the reviewer for helping us to clarify this point. In the results section, we now write:

"Tumors are not found in the immediate vicinity of archetypes for statistical reasons (Supplementary text)."

In the supplementary material, we now write:

"The observation that tumors are not found in the immediate vicinity of archetypes is an artifact of visualizing high-dimensional data.

In examining polyhedra of dimension three or higher (4 archetypes or more), we can observe that there are no tumors in the direct vicinity of the archetypes. This observation has a statistical explanation: measurement error tends to push archetypes away from the data. In addition, projecting higher dimensional polyhedra onto a plane – a step necessary for visualization – artificially increases the density of generalists compared to specialists: the more archetypes, the higher the dimension, and the smaller the density of projected data close to the archetypes.

To illustrate this, we simulate data points by sampling uniformly from polyhedra with 3, 4, 5 and 6 archetypes. After adding measurement noise, we infer the position of archetypes using ParTI and project the data onto the face of the polyhedron defined by the first 3 archetypes.

As the panel with 3 archetypes shows, adding noise pushes the inferred archetypes away from the data. Upon orthogonally projecting the data onto a face of the polyhedron, archetypes that are not part of the face increase the projected density close to the middle of the face and thus decrease density in the vicinity of archetypes.

This decreased projected density in the vicinity of archetypes is observed even though density in the high-dimensional polyhedron is uniform. The more archetypes, the smaller the density in the vicinity of archetypes after projecting the data.”

We now also add a figure showing the density of tumors of the faces of the 5-archetype polyhedron (Fig. S2B).

3. Pg 4. There may be multiple gene expression routes to the archetypes. How does the method handle this? Would the signal be diluted as the number of routes increases? Could this explain the 'clouds' in expression space?

We thank the reviewer for raising this important point. In the discussion, we now write:

“Given the genetic diversity of cancer, it is interesting to ask how multi-task evolution handles the possibility that multiple gene expression profiles may maximize performance at a task. If different gene expression profiles optimize performance at the same task, there will be multiple archetypes for this task, each with its own gene expression profile. We do not observe this: the archetypes we find clearly differ in their tasks. However, comparing genes for a given task in different tumor types reveals that tasks have a tissue-specific flavor (Database S6).”

4. Pg 4. How much does tumor purity dictate how sharply the tumor can be resolved as archetypes?

We have now repeated our analysis on pure and impure glioma tumors separately, defined as tumors of the top and bottom quantiles of the ESTIMATE score (Yoshihara et al., *Nat Comm* 2013). With impure tumors, we find that same three archetypes as when using all tumors. With pure tumors, we find two archetypes which are the archetypes closest to pure tumors in the full analysis. Thus, our analysis seems to apply to both pure and impure tumors.

We now added 4 figures to illustrate these results. In the results section, we now write: “The identified archetypes are robust to inter-tumor variations in purity and clonal heterogeneity (Fig. S1D-H, Supplementary Text).”

In the supplementary material, we now write:

“To determine how tumor purity affects ParTI's ability to resolve archetypes, we stratify the analysis according to purity of low-grade glioma tumors. Applying ParTI to the 25% low-grade glioma tumors with high purity reveals two archetypes. Applying ParTI to the 25% low-grade glioma tumors of lowest purity reveals three archetypes. While a polyhedron with three archetypes is poor fit to the data, archetypes identified from low purity gliomas match archetypes identified from all glioma (see panel G of Fig. S1). Principal component analysis comparison of archetypes identified from all gliomas, pure gliomas and impure gliomas shows that the same archetypes are inferred from tumors of different purities. The two archetypes identified from pure tumors match glioma archetypes 2 and 3 whereas all three glioma archetypes are identified from low-purity tumors. These results suggest that ParTI identifies archetypes relevant to a given tumor subset.”

5. Pg 5. *What is the relationship between clonal diversity at the mutational level and the ability to resolve sharply defined archetypes?*

To investigate the relationship between clonal diversity and the ability to resolve archetypes, we now apply ParTI separately on heterogeneous and homogeneous metabric breast tumors. We find that heterogeneous tumors have 4 archetypes which are the same as the archetypes inferred from all tumors (cell division, tissue remodeling & invasion, biomass & energy, Her2). Three archetypes are found in homogeneous tumors, two of which are shared with heterogeneous tumors (biomass&energy, Her2). The third archetype of homogeneous tumors is the immune interaction. These findings suggest that archetypes can be resolved from both clonally homogeneous and heterogeneous tumors, with most tasks shared among these tumors, and specific tasks relevant to certain tumors only.

We added a figure to illustrate this point (Fig. S1H). In the results, we now write:

“The identified archetypes are robust to inter-tumor variations in purity and clonal heterogeneity (Fig. S1D-H, Supplementary Text).”

In the supplementary material, we now write:

“To determine how the inferred tasks are influenced by clonal heterogeneity, we stratified our analysis according to the Mutant-Allele Tumor Heterogeneity Score (MATH), an established measure of clonal heterogeneity. The MATH score is defined as the median absolute deviation of the frequency of mutations found in a tumor. In homogeneous tumors, cancer cells share the same alleles so that the frequency of different mutations tends to be similar and the MATH score is low. The MATH score is high when different mutations occur at different frequencies, as happens in heterogeneous tumors in which multiple lineages of subpopulations of cancer cells coexist.

Analyzing the 25% tumors with highest MATH score in the Metabric cohort, we find the same four archetypes as when analyzing all Metabric tumors together: 1. cell division, 2. tissue remodeling & invasion, 3. biomass & energy, 4. Her2. Analyzing Metabric tumors in the lowest MATH score quartile, we find three archetypes. Two archetypes are shared with heterogeneous tumors (1. Her2, 2. biomass&energy). The third archetype corresponds to the task of immune interaction.

Thus, ParTI can be applied to both clonally homogeneous and heterogeneous tumors. Most tasks appear to be shared among tumors, while certain tasks are seen when focusing on tumors with specific properties – here homogeneous tumors.”

6. Pg 5. *how do the archetypes map to known cancer subtypes (iclusters in Metabric or PAM50 in TCGA breast cancer)?*

We now report statistically significant associations between archetypes and breast cancer subtypes in Table 1.

Significant associations between archetypes and integrative clusters are reported in Table S2. For example, in the metabric breast cancer cohort, Integrative Cluster 10 tumors, Pam50 basal tumors and triple negative tumors are found close to the cell division archetype (archetype 1). Integrative Cluster 4 and the normal Pam50 subtype are over-represented in tumors closest to archetype 2 (invasion & tissue remodeling). Archetype 3 (biomass&energy) is enriched with tumors from the LumB Pam50 subtype while the Her2 Pam50 tumors are found close to the archetype 4 (Her2).

In other TCGA cancer types, associations between archetypes and cancer subtypes are weaker, perhaps due to the 5-fold smaller sample size. We report significant associations between cancer subtypes and archetypes in Table S2.

7. Pg 6. 2nd last paragraph. Available single cell data from various cancer types could resolve this convincingly.

We now address how single-cell intra-tumor heterogeneity supports inter-tumor diversity in two new paragraphs and in the new Figure 4. Page 9-11 now reads:

“We considered so far diversity in gene-expression between different tumors. We now ask about how this inter-tumor variation relates to the variation between single cancer cells inside a tumor (Tirosh *et al.*, 2016; Kim *et al.*, 2018). One possibility is that each tumor is composed of a combination of specialist cells, that specialize in the different cancer tasks. Each specialist cell thus has an expression program at one of the archetypes. The position of the entire tumor in the polytope is determined by the ratios of these specialist cell types.

This possibility is inconsistent with the data: if tumors were made of specialist cells, specialist tumors should be nearly most homogeneous in a single type of specialist cells, whereas generalist tumors should be heterogeneous. We tested this by considering heterogeneity in each tumor according to a mutation-based score. We find that specialist breast tumors can be either homogeneous or heterogeneous (Fig. S4A), and that tumors that specialize in the cell division task are more heterogeneous than generalist tumors ($p=1.6 \times 10^{-5}$, Fig. S4A).

We therefore asked whether single cancer cells within a tumor span a continuum of gene expression. Projecting single cancer cell gene-expression data onto the space defined by the first three principal components of breast tumor gene expression confirms that single cells from individual tumors can be specialist or generalists at the different tasks (Fig. 4A).

Finally, we used a concept from ecology (Schluter, 1996; Sheftel *et al.*, 2018) to hypothesize that the variation between individual cells in a tumor will fall along the subspace defined by variation between different tumors. The reason for this is that the evolution of cells within a given tumor occurs under tradeoffs between the same tasks that apply to all tumors (Shoval *et al.*, 2012). The ecological analogy, called “evolution along lines of least genetic resistance”, is that variation within given species in traits falls along the directions defined by variation between species.

To test this, we analyzed single-cell transcriptomic data from breast tumors (Karaayvaz *et al.*, 2018). We compared the fraction of the variance in single cancer cell gene expression explained by the first 5 principal components (PCs) of the single cell data to the variance in single cancer cell gene expression explained by the first 5 PCs of Metabric breast tumors (Curtis *et al.*, 2012).

If Metabric tumors and single cancer cells vary along the same directions in gene expression space, PCs computed on Metabric tumors will explain as much variation in single cancer cell gene expression as PCs computed on the single cell data. On the other hand, if single cell gene expression varies along directions different from Metabric tumors, PCs computed on the Metabric data will explain a negligible fraction of the variation in single cell gene expression because the space of gene-expression has thousands of dimensions along which tumors and cells could vary.

We find that the first 5 Metabric tumor PCs explain 25.4% of the variation explained by the first 5 single cancer cells PCs, with little change if different numbers of PCs are used between 5 to 50 (Fig. 4, Fig. S4B-C). This alignment between inter-tumor and intra-tumor variation is much higher than expected by chance: the same number of random directions in gene expression

space only explains 1.5% of intra-tumor variation ($p < 0.001$, Fig. 4). The remaining ~75% of the variation may be due to stochasticity in single-cell gene expression, measurement error and neutral variation.”

8. *Pg 8. “Mutations in TP53 in breast cancer...”. This is association and not causation. There could easily be confounding effects that yield the association such as known subtypes enriched for specific driver mutations. Thus the association may relate to a cell type of origin better able to tolerate TP53 mutation.*

We thank the reviewer for bringing up this caveat. In the discussion, we now write: “There could be alternative interpretation to the observed associations between cancer tasks and driver mutations. For example, different cell type of origin may have different propensity to specialize in different tasks and acquire different mutations. At the moment, this specific explanation has limited support, at least in breast cancer: most breast cancers are thought to originate from luminal progenitors, so that most inter-tumor diversity cannot explained by the cell of origin, except in rare tumors (claudin low). Nevertheless, clarifying the origin of the observed association between driver mutations and cancer tasks could be the object of future research, perhaps by comparing gene expression in genetically engineered mouse models with different driver mutations.”

9. *That the cell lines specializing in specific tasks are more sensitive to the drugs that inhibit these tasks is not particularly surprising? It is not clear how this particular analysis is better able to resolve this than traditional pathway analysis. Could the authors elaborate?*

We thank the reviewer for giving us this opportunity to clarify the purpose of the drug sensitivity analysis in our study. We did not perform drug sensitivity analysis to show that archetypes are a better way to predict drug sensitivity than pathway analysis. Instead, the purpose of the drug sensitivity analysis is to test the prediction that task specialists should be sensitive to drugs that interfere with that task. Examining drug sensitivities in the context of tasks and trade-offs has the benefit of identifying order and structure across diverse cancer datasets – gene expression, clinical properties, driver genetic alterations – and propose an explanation of this structure based on evolutionary theory.

We now clarify our purpose in the main text, page 7: “To further test the hypothesis that tumors face trade-offs between conflicting tasks, we sought a way to test whether tumors near an archetype depend on its task more than other tumors. For this purpose, we employ the drug sensitivity of different tumors, reasoning that tumors near an archetype will be most sensitive to drugs which specifically disrupt that task.”

10. *The paper lacks any benchmarking data. What are the appropriate alternative approaches to this type of inference? As the paper has a methodological component, a comparative component to complementary methods should be included.*

We agree that benchmarking helps readers grasp the strengths and weaknesses of different methods. But we cannot do benchmarking in the present study because we are not aware of alternative methods to achieve our aim: explain why tumors differ from each other based on selection trade-offs and infer these trade-offs from large-scale data. Other analysis methods have different goals, so that they cannot be compared on a common metric. This prevents us from doing benchmarking here.

11. *Is the source code for the model and analysis available in a public repository (e.g. github?)*

We now include a code availability statement in the main text:

“Code availability Input data files as well as scripts to reproduce the analyses can be downloaded at: <https://data.mendeley.com/datasets/2r9h9xwzm3/draft?a=8dd0167f-fc1c-463b-8594-28e14ff094a7>”

REVIEWERS' COMMENTS:

Reviewer #2 (Remarks to the Author):

The authors have given thoughtful responses to my comments and I have no further concerns.